# Evaluation of an integrated intervention to reduce psychological distress and intimate partner violence in refugees: Results from the Nguvu cluster randomized feasibility trial

M. Claire Greene[1]*, Samuel Likindikoki[2], Susan Rees[3], Annie Bonz[4], Debra Kaysen[5], Lusia Misinzo[2], Tasiana Njau[2], Shangwe Kiluwa[2], Rachael Turner[6], Peter Ventevogel[7], Jessie K. K. Mbwambo[2], Wietse A. Tol[8,9,10]

1 Program on Forced Migration and Health, Columbia University Mailman School of Public Health, New York, New York, United States of America, 2 Department of Psychiatry, Muhimbili University of Health and Allied Sciences, Dar es Salaam, Tanzania, 3 School of Psychiatry, University of New South Wales, Sydney, New South Wales, Australia, 4 HIAS, Silver Spring, Maryland, United States of America, 5 Department Psychiatry, Public Mental Health and Population Sciences, Stanford University, Palo Alto, California, United States of America, 6 Department of Community-Public Health, Johns Hopkins School of Nursing, Baltimore, Maryland, United States of America, 7 Public Health Section, United Nations High Commissioner for Refugees, Geneva, Switzerland, 8 Department of Mental Health, Johns Hopkins Bloomberg School of Public Health, Baltimore, MD, United States of America, 9 Department of Public Health, Global Health Section, University of Copenhagen, Copenhagen, Denmark, 10 Peter C. Alderman Foundation, HealthRight International, New York, New York, United States of America

* mg4069@cumc.columbia.edu

**Data Availability Statement:** The data contain sensitive information in a vulnerable population. Given concerns about potential risks to human

## Abstract

### Introduction

The complex relationship between intimate partner violence and psychological distress warrants an integrated intervention approach. In this study we examined the relevance, acceptability, and feasibility of evaluating a multi-sectoral integrated violence- and mental health-focused intervention (Nguvu).

### Methods

We enrolled 311 Congolese refugee women from Nyarugusu refugee camp in Tanzania with past-year intimate partner violence and elevated psychological distress in a feasibility cluster randomized trial. Women were recruited from local women's groups that were randomized to the Nguvu intervention or usual care. Participants from women's groups randomized to Nguvu received 8 weekly sessions delivered by lay refugee incentive workers. Psychological distress, intimate partner violence, other wellbeing, and process indicators were assessed at baseline and 9-weeks post-enrollment to evaluate relevance, acceptability, and feasibility of implementing and evaluating Nguvu in refugee contexts.

### Results

We found that Nguvu was relevant to the needs of refugee women affected by intimate partner violence. We found reductions in some indicators of psychological distress, but did not

subjects, the Institutional Review Boards at the Muhimbili University of Health and Allied Sciences, and the Tanzania National Institute for Medical Research have recommended that all data requests are reviewed and obtain adequate approvals for secondary data analysis to protect the study participants. The data underlying these results will be made available upon request to the University of Copenhagen Research Data Management center (data@sund.ku.dk). We will require that individuals requesting permission to use these data receive approval from their home institution's Institutional Review Board for secondary data analysis prior to receiving the data in effort to protect the safety of research participants.

**Funding:** The research was funded by an award issued to WAT from the Research for Health in Humanitarian Crises (R2HC) initiative, co-funded by the Department for International Development (DFID) and the Wellcome Trust, overseen by Elrha (https://www.elrha.org/programme/research-for-health-in-humanitarian-crises/). The funders had no role in study design, data collection and analysis, decision to publish, or preparation of the manuscript.

**Competing interests:** The authors have declared that no competing interests exist.

identify sizeable changes in partner violence over time. Overall, we found that Nguvu was acceptable and feasible. However, challenges to the research protocol included baseline imbalances between study conditions, differential intervention completion related to intimate partner violence histories, differences between Nguvu groups and facilitators, and some indication that Nguvu may be less beneficial for participants with more severe intimate partner violence profiles.

## Conclusions

We found evidence supporting the relevance of Nguvu to refugee women affected by partner violence and psychological distress and moderate evidence supporting the acceptability and feasibility of evaluating and implementing this intervention in a complex refugee setting. A definitive cluster randomized trial requires further adaptations for recruitment and eligibility screening, randomization, and retention.

## Trial registration

ISRCTN65771265, June 27, 2016.

## Introduction

By the end of 2018, the number of displaced persons globally exceeded 70 million, levels unseen since the Second World War, with approximately 26 million people who were forced to flee their country due to war, violence, or persecution [1]. Recent estimates from the World Health Organization suggest that 22.1% of individuals in conflict-affected populations has a mental disorder at any point in time, which is approximately three times higher than non-conflict affected populations [2]. The burden of common mental disorder is estimated to be even greater among refugees and populations displaced by emergencies, but reliable epidemiological estimates are limited [3].

Intimate partner violence (IPV) is the most common form of gender-based violence, also in humanitarian settings. Regionally, studies estimate that 39–44% of women in sub-Saharan Africa have experienced IPV in their lifetime [4,5]. Research conducted among refugees in sub-Saharan Africa consistently identify an association between IPV and mental health problems [6–8]. The risk for IPV and psychological distress is elevated in refugee and conflict-affected populations due the presence of unique and shared risk factors, chronic adversity, and limited prevention and treatment services [2,9–12]. Emerging longitudinal research suggests that IPV and psychological distress may reinforce each other, thus resulting in a vicious cycle whereby IPV increases risk for psychological distress, and psychological distress may put women at further risk of recurrent and more severe IPV [13–17]. These findings should be interpreted using a critical feminist and human rights analysis to avoid pathologizing and blaming women for the crime of male-perpetrated violence in relationships [18,19]. The mechanism underlying the relationship between psychological distress and future IPV risk and severity has not been well described, however it is possible that women may be targeted and victimized when they are less able to protect themselves due to impaired functioning. Further, women with common mental health conditions related to IPV (e.g, depression, anxiety, post-traumatic stress disorder) may also suffer from low self-esteem, self-blame, poverty and alcohol or drug use, factors that may reduce their capacity to escape from future victimization. It is

possible that improvements in mental health may lead to better functioning and subsequent utilization of safety and psychosocial support services. The complex relationship between IPV and mental health therefore indicates the need for an integrated response to effectively address the dual priorities for improved mental health and reduction in IPV in global public health and human rights.

Nguvu, Kiswahili for strength and power, is an integrated intervention that was developed among female refugees from the eastern Democratic Republic of the Congo (DRC) to reduce IPV and psychological distress. Research among women in the eastern DRC and refugees from this region consistently reports high levels of gender-based violence, particularly IPV (e.g., 31% of adult women in the eastern DRC), as well as co-occurring mental health problems in survivors of IPV [8,20]. However, the reliability and availability of prevalence estimates among refugee populations specifically are limited [20]. One population-based survey conducted in the eastern DRC found that experiencing IPV is associated with a higher probability of common mental health problems including depression (64.9% vs. 31.0%), post-traumatic stress disorder (77.2% vs. 43.9%), and suicidal ideation (42.4% vs. 20.5%) or attempts (33.1% vs. 9.7%) [21].

It is important to initiate operational research to bridge the gap that often exists in humanitarian settings between mental health interventions that are usually not tailored to the needs of IPV survivors and routine IPV interventions that often lack the clinical competency to effectively address co-occurring mental health issues [22]. Rigorous research on strategies for reducing IPV among women experiencing ongoing violence in refugee settings and humanitarian emergencies is limited [20]. In this study we aim to conduct a cluster randomized controlled feasibility trial to evaluate the relevance, acceptability, and feasibility of the Nguvu intervention and effectiveness evaluation procedures in a complex refugee setting. Results from the preliminary intervention cohort study revealed unique implementation considerations in refugee contexts relative to other low-resource settings that may impede the feasibility of implementing and evaluating the Nguvu intervention with adequate fidelity and scientific rigor. These considerations include: 1) differences in the study context (e.g., instability and high rates of in- and out-migration resulting in challenges retaining participants in long interventions), 2) health providers (e.g., task-shifting with refugee incentive workers to leverage knowledge and trust, while overcoming the limited human resource capacity to provide mental health and IPV response services), 3) the multi-sectoral nature of the intervention positioned within humanitarian health and protection systems that have different mandates and are managed by different implementing agencies, 4) challenges with communication and coordination across sectors, and 5) other challenges with the broader service delivery system (e.g., high rates of staff turnover) [23]. Results from this study are intended to inform the design and conduct of a definitive cluster randomized effectiveness trial.

## Materials and methods

### Population

This study was conducted in Nyarugusu refugee camp in northwestern Tanzania between April and September 2017. Nyarugusu refugee camp was established in 1996 in response to over 150,000 Congolese fleeing conflict in the eastern provinces of the DRC. As of early 2015 there were over 60,000 refugees in Nyarugusu refugee camp, most of whom were Congolese. Beginning in April 2015 there has been a large influx of refugees arriving from Burundi and additional arrivals from the DRC leading to a population of over 150,000 refugees in Nyarugusu as of 2018, about half of whom are Congolese [24]. During this feasibility trial there were ongoing resettlement efforts focused on Congolese refugees.

## Procedures

We employed a cluster randomized feasibility trial design in order to build upon the existing infrastructure of local women's groups (i.e., clusters), which are organized at the village-level by the International Rescue Committee [23]. These women's groups are aimed at strengthening social networks and providing skills training. Women's groups focused on a variety of skills and objectives including cooking, village savings and loans, weaving, etc. Given the strength of these networks and the potential for contamination within women's groups, we elected to randomize women's groups as clusters as opposed to randomizing eligible women individually. Women's groups were allocated to the Nguvu intervention versus the usual care condition using a simple, unrestricted randomization approach with approximately equal allocation conducted by an investigator not affiliated with the current study [25]. The random number sequence was generated in Stata, Version 14 [26]. All clusters were randomized at the same time thus reducing concerns about allocation concealment [27].

We obtained a list of women's groups from the International Rescue Committee and found that 43 of the 63 listed women's groups were operational. In these 43 women's groups there were 647 members. We approached the leaders of the 43 women's groups to receive their permission to deliver a brief summary of the study and invite interested members to participate. The study was presented to the women's group members as a study of women's health and wellbeing to avoid interested women being identified by their peers as IPV survivors. All 43 operational clusters were randomized and agreed to voluntary recruitment, screening and enrollment of their members.

Eligible participants were adult (18+ years) female Congolese refugees living in Nyarugusu refugee camp who were married or in a relationship in the last 12 months and reported past-year physical or sexual IPV as well as elevated levels of psychological distress. Past-year physical or sexual IPV was identified by affirmative responses to at least one of the following items from the Abuse Assessment Screen [28]: 1) Within the last 12 months have you been hit, slapped, kicked, or otherwise physically hurt by an intimate partner?; or 2) Within the last 12 months has an intimate partner forced you to have sexual activities? Elevated psychological distress was operationalized as an average item score greater than or equal to 1.75 on the Hopkins Symptom Checklist and/or greater than 1.00 on the Harvard Trauma Questionnaire when the item responses on both of these measures were scored on Likert scales from zero to three [29,30]. These cutoffs were based on prior research with Congolese women for whom these scores indicated significant psychological distress [31]. We excluded women at imminent risk of suicide or observable signs of severe psychiatric disorder that would impede participation in intervention sessions. Women excluded due to high risk of suicide were linked immediately with a counselor as per our safety protocol.

Women recruited for the study were allocated to study condition based on the randomization of women's groups (i.e., clusters). Women recruited from women's groups randomized to Nguvu were enrolled in the intervention. Women who were recruited from women's groups randomized to the control condition received information about available protection and mental health services in Nyarugusu. We planned to enroll 400 participants in this study, which would allow us to identify a small to moderate effect size for depression/anxiety measured using the average item score on the Hopkins Symptom Checklist (mean difference = 1.6) and post-traumatic stress symptoms measured using the average item score on the Harvard Trauma Questionnaire (mean difference = 1.3) [29,30]. Enrolling 200 participants per study condition from the 63 original women's groups with approximately equal allocation to each study condition was necessary to achieve at least 80% power to detect a small to moderate effect of the intervention on mental health outcomes accounting for up to 20% attrition and an

intraclass correlation ranging from 0.1–0.5 within subjects and 0.1–0.3 within cluster [32]. Parameter estimates used to inform this power calculation were based on estimates from a previous cluster randomized controlled trial of Cognitive Processing Therapy conducted in the eastern Democratic Republic of the Congo that used the same outcome measures as were used in this feasibility trial [31].

All eligible women who provided written informed consent were invited for a baseline assessment. All participants completed an endline assessment at nine weeks post-enrollment, which corresponded to one week after the intervention for women who were assigned to the Nguvu study condition. A group of research assistants not involved in provision of health or protection services completed the recruitment and assessment procedures. All research assistants were female Congolese refugee incentive workers who completed a 10-day training in research study procedures. Participants and intervention facilitators were not masked to study allocation. While research assistants were not informed of the participants' allocation, it may have become apparent through information shared during the endline assessment.

All procedures were reviewed and approved by the Johns Hopkins Bloomberg School of Public Health Institutional Review Board (IRB0007219), the Muhimbili University of Health and Allied Sciences Institutional Review Board (2014-10-27/AEC/Vol.X/56), and the Tanzania National Institute for Medical Research (NIMR/HQ/R.8a/Vol.IX/2016). The Johns Hopkins Institutional Review Board protocol is provided in S2 File.

## Intervention and usual care conditions

The Nguvu intervention was developed in an effort to integrate evidence-based intervention approaches to reduce psychological distress and IPV in low-resource, refugee settings [32]. The Nguvu intervention consisted of one individual session provided by a facilitator followed by seven weekly group sessions delivered in person by a pair of facilitators to groups ranging in size from 6–13 women. Nguvu group sessions began when at least 6 women were enrolled and allocated to a given group. Intervention facilitators were lay refugee incentive workers in Nyarugusu already working with the humanitarian partner (International Rescue Committee) who had some experience working with protection and psychosocial support programs in the camp and had received training from experts in trauma-informed psychological and gender-based violence interventions. Incentive workers are refugees who undertake work related to the provision of humanitarian assistance and receive fixed compensation referred to as an 'incentive' [33]. Consistent with task-shifting approaches [34], these incentive workers were non-specialists and had no prior experience in implementing psychological interventions beyond the basic psychosocial support programs offered by the humanitarian partner. The Nguvu intervention integrated elements of Advocacy Counseling and Cognitive Processing Therapy. Details of the intervention are provided in the intervention development study [23,32].

To contextualize and strengthen the relevance of this intervention within a protracted refugee setting, we conducted qualitative free listing and key informant interviews with Congolese refugee incentive workers living in Nyarugusu Refugee Camp, which is located in northwestern Tanzania [35]. The free listing and key informant interviews with refugee incentive workers confirmed the persistence of IPV among women in their community as well as the following mental health problems that commonly affected IPV survivors: *msongo wa mawazo* (stress, too many thoughts), *huzuni* (deep sadness), and *hofu* (fear). Using information from these qualitative interviews along with expert consultation and a desk review [8], we utilized a modified six-session version of Cognitive Processing Therapy (CPT), an evidence-based intervention developed for survivors of assault [36,37]. The twelve-session version of CPT has been

shown to reduce mental health problems among Congolese survivors of gender-based violence in the DRC [31]. In the Nguvu intervention, CPT was combined with two sessions of advocacy counseling, which has demonstrated some reductions in IPV victimization among women [38–40]. We hypothesized that key mediators of this integrated intervention approach would include improved social support, increased coping, and support seeking [32]. Details of the development and preliminary testing of the intervention through a non-controlled intervention cohort study that was previously conducted to pilot test the intervention and assess the psychometric properties of the study outcome measures are reported elsewhere [23].

Women recruited from women's groups randomized to the usual care condition received information about existing services for mental health and protection from the research assistant at the end of the baseline assessment. The gender-based violence response program that existed at the time of the study consisted of case management (including basic counseling) and referrals to protection, medical, or legal services. These services included legal consultation and aid services, education about women's rights, and arranging safe shelter and accommodations [41]. Women in the Nguvu study condition were also able to access usual care services available in Nyarugusu refugee camp, including those for women seeking support for IPV.

## Measures

The primary outcomes measured in this study were psychological distress and IPV. Psychological distress was measured using the Hopkins Symptom Checklist (HSCL-25) [29], which measures symptoms of anxiety and depression, and the Harvard Trauma Questionnaire (HTQ) [30], which measures symptoms of post-traumatic stress. These scales assess symptoms that were consistent with the descriptions of the priority mental health problems among IPV survivors that were reported in our formative qualitative research. At screening, physical and sexual violence were assessed using the Abuse Assessment Screen [28]. Psychological, physical, and sexual IPV were measured in the baseline and follow-up assessment using an adapted version of the Conflict Tactics Scales that was developed for the World Health Organization multi-country study on women's health and domestic violence against women and the Demographic and Health Surveys [42–44]. The 11 Demographic and Health Survey items that were used include questions assessing lifetime and past two-week psychological violence (2 items: humiliate, threaten), physical violence (7 items: push/shake/throw, slap/twist arm/pull hair, punch, kick/drag, strangle/burn, threaten with weapon, attack with weapon), and sexual violence (2 items: forced sex, forced other sexual acts). IPV frequency was calculated as the mean reported frequency of each type of IPV (physical, psychological, sexual) over the past two-weeks.

We measured history of potentially traumatic events using the HTQ [30] and included measures developed in the eastern DRC to assess functional impairment, coping and service use, and social support [31,45]. In addition to these outcome measures, we assessed demographic characteristics of the sample. Given ongoing resettlement efforts at the time of the study we assessed participant preferences toward resettlement and whether they had begun the process.

All measures were translated into Kiswahili, adapted, and validated in the intervention cohort study, which was conducted among women recruited using the same eligibility criteria as the current study within a zone in Nyarugusu that was not included in this feasibility trial [23]. This intervention cohort study found good test-retest reliability, inter-rater reliability, internal consistency, and external construct validity for most measures. The exceptions were poor test-retest reliability for the sexual violence IPV subscale and low internal consistency of the functional impairment measure. We therefore made adaptations for the current study by adding a script that was read by the interviewer prior to administering the sexual violence

items to acknowledge the sensitivity of these questions, while reassuring the participant that her answers would be kept confidential and there would not be any consequences of her reporting these experiences [23]. In addition, ten of the 22 items on the functional impairment measure were removed based on their low inter-item correlation and lack of face validity for a refugee context, which improved test-retest reliability of the measure [23].

## Statistical analyses

**Relevance.** As shown in the study flow diagram, we calculated the proportion of women screened who were eligible and enrolled in Nguvu as well as reasons for exclusion. We examined the distribution of demographic and wellbeing (e.g., psychological distress, IPV, functioning) characteristics in the full sample at baseline and reported their mean and standard deviation or sample size and proportion.

We examined sensitivity to change in IPV, psychological distress, and functional impairment measures by calculating the mean change in primary and secondary outcomes from baseline to follow-up, the effect size, as well as the Pearson correlation between these change scores. This analysis was conducted to determine whether these assessment tools would be suitable as primary outcome measures in a fully powered, definitive randomized controlled trial.

To examine whether we were able to detect between-group differences we estimated the difference in means between the Nguvu and usual care condition at the post-intervention assessment. These intention-to-treat, complete case analyses included participants who completed the follow-up assessment. Analyses compared outcomes between study conditions as they were assigned at randomization. Mixed effects models included random intercepts for women's group to account for clustering. We also examined the sensitivity of our inferences to further adjustment in these mixed effects models by including: 1) demographic imbalances between study conditions at baseline, and 2) baseline levels of primary outcomes and demographic imbalances between study conditions. We then conducted a per protocol analysis of completers (i.e., those who attended 6 or more sessions) compared to usual care on mental health and IPV post-intervention differences controlling for baseline levels of primary outcomes and demographic imbalances between study conditions using a mixed effects model.

We conducted an exploratory analysis of moderators of psychological distress and IPV frequency outcomes as an early indication of subgroups that might be particularly responsive to the intervention. To reduce the number of tests in this exploratory analysis we used composite outcomes for psychological distress, which was the sum of depression, anxiety, and PTSD symptoms, and IPV frequency, which was the average of the psychological, physical, and sexual IPV frequency measures. Moderators included baseline levels of primary outcomes, number of potentially traumatic events experienced, and marital status. Moderators were recoded as high vs. low based on median splits of the full distribution of the baseline value. We constructed mixed effects models stratified by levels of baseline moderators and continued to account for clustering by women's group using a random intercept.

**Acceptability.** We examined baseline correlates of intervention completion, which we defined as six or more sessions, as was done in the intervention cohort study, among Nguvu participants using logistic regression models. Correlates we tested included age, literacy, education, religion, marital status, household composition, length of time living in Nyarugusu, resettlement status, psychological distress, IPV, functioning, study condition, and which Nguvu group and intervention facilitators they were assigned to (if in the Nguvu condition). To examine safety of the intervention, an indicator of acceptability, we documented adverse events.

**Feasibility.** We examined the distribution of demographic and wellbeing characteristics comparing the Nguvu and usual care participants at baseline to assess whether cluster randomization led to balanced study conditions. We explored baseline correlates of study attrition in the full sample using logistic regression models to calculate odds ratios and 95% confidence intervals. Baseline variables were the same as those included in the analysis of correlates of intervention completion (see *Acceptability*). To evaluate feasibility of conducting a definitive randomized trial of Nguvu we monitored and described any deviations to the study protocol, including any changes to implementation of the recruitment, screening, assessment, or intervention procedures.

All analyses were conducted in Stata, Version 14 [26].

## Results

### Sample characteristics at baseline

Of the 647 women registered with 43 local women's groups (i.e., sampling frame) we screened 401 women from the 43 operational women's groups, of which 311 women were enrolled. The most common reasons that individuals who were screened were ineligible were that they did not report past-year IPV (n = 60), they did not report moderate or severe psychological distress (n = 23), had not been married or in a relationship in the past year (n = 13), and one participant reported imminent risk of suicide and was linked immediately with a counselor as per our safety protocol. Two potential study participants who may have been eligible were not enrolled because one screening interview was not completed and eligibility of another potential participant was inaccurately assessed by study staff. Randomization of 21 women's groups to the Nguvu study condition and 22 to the usual care condition resulted in 158 enrolled women being allocated to intervention and 153 women allocated to usual care (n = 311 total; Fig 1). Except for one of the women's groups assigned to the Nguvu condition, at least one woman from all groups was enrolled in the study (median number of women enrolled per women's group = 6; IQR = 3,10).

At baseline participants were 33.5 (SD = 9.0) years of age and had lived in Nyarugusu refugee camp for 17.6 (SD = 4.8) years on average. Most participants were married and living with their partner (73.6%), Wabembe ethnicity (73.6%), literate (73.7%), and Christian (18.7% Catholic, 18.7% Methodist/Free/United, 52.3% other Christian denomination). About one-third had a secondary school education or higher. Almost all participants had children (96.7%; 4.8 children, on average). The average household size was 7.3 (SD = 3.3) people. Almost all participants preferred to be resettled, but approximately one-third had begun the process. Few reported completing later stages of the resettlement process (e.g., 4.9% completed a resettlement health screen, 3.3% had participated in cultural orientation sessions). At the cluster level, the distribution of these characteristics differed across conditions. The proportion of women who were not married, but living with their partner, more highly educated, and literate was higher in women's groups allocated to the Nguvu relative to usual care condition. The mean number of children appeared higher in clusters allocated to usual care. At the individual level, the distribution of these demographic characteristics was similar across participants allocated to the Nguvu and usual care conditions with few exceptions. Participants in the Nguvu condition were more likely to have a secondary education, less likely to report their religious affiliation as Methodist/Free/United, and had fewer children, on average (Table 1).

### Mental health and IPV at baseline (relevance)

All findings are briefly summarized in Table 2. Participants reported elevated psychological distress as indicated by a greater than moderate amount of depressive, anxiety, and post-

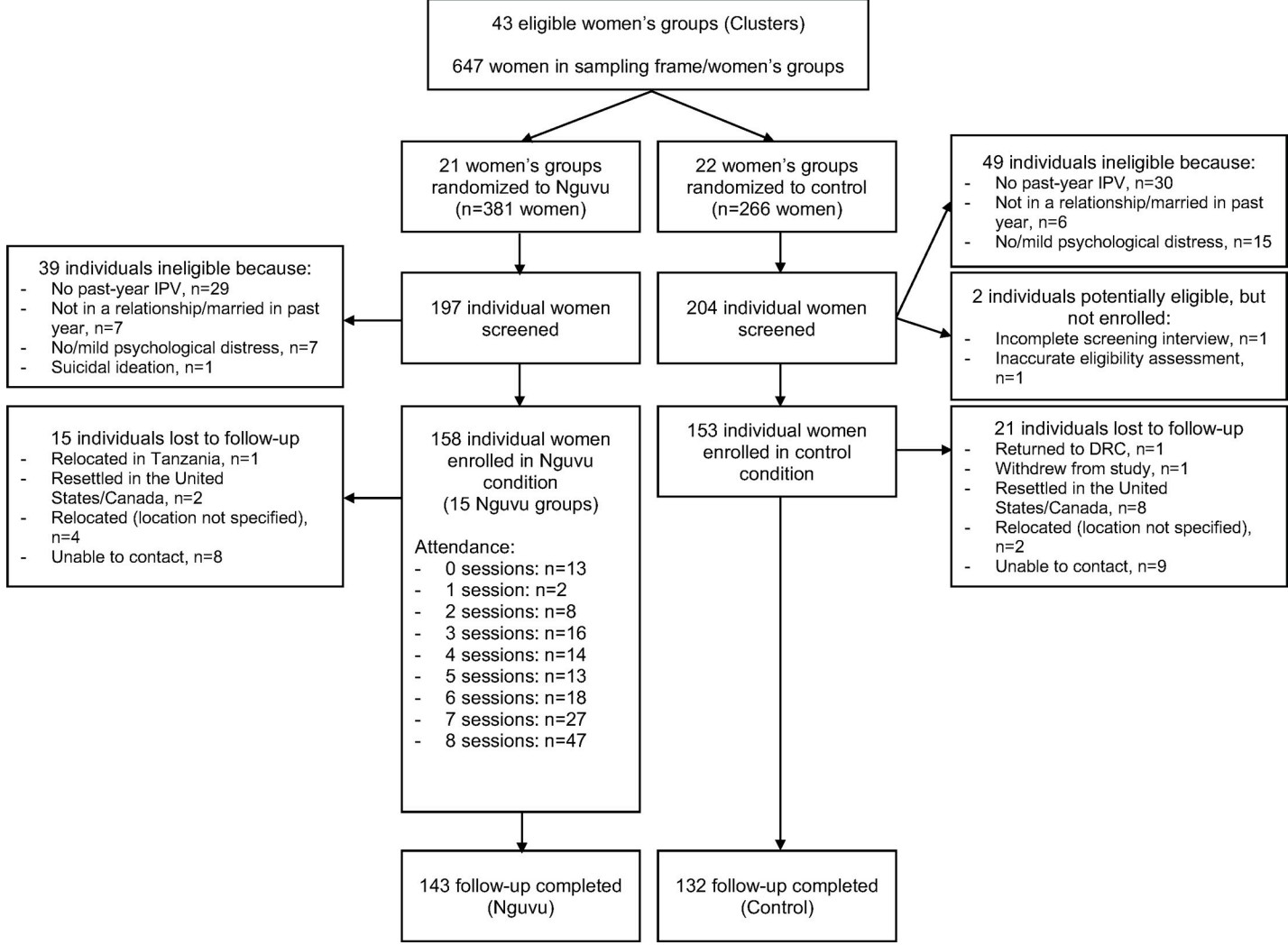

**Fig 1. Flow of participants.** Flow diagram of cluster randomization and allocation, participant screening, enrollment (n = 311), and follow-up (n = 275). The reasons for exclusion at the stage of screening do not sum to the total number of people who were ineligible because some participants reported more than one exclusion criterion.

traumatic stress symptoms. All primary outcome measures displayed good internal consistency at baseline (Anxiety: $\alpha$ = 0.761, Depression: $\alpha$ = 0.741, PTSD: $\alpha$ = 0.722). These symptoms were higher among the usual care participants relative to participants in the Nguvu study condition. Functioning was similar between groups and reflected a little to moderate amount of difficulty completing common life tasks ($\alpha$ = 0.841. Almost all participants reported experiencing controlling behaviors and IPV-related injury (i.e., bruising, aches, injury, broken bone, or doctor visit due to IPV) in the past two weeks. The majority of participants reported experiencing psychological IPV (67.2%), physical IPV (82.3%), and/or sexual IPV (95.2%) perpetrated by their current or most recent partner. More than half (57.9%) of participants reported experiencing all forms of IPV perpetrated by their current or most recent partner and it was rare that participants experienced only one form of IPV. The frequency of IPV in the past two weeks was positively skewed for all forms of IPV. Mean days experiencing a specific form of IPV in the past two weeks was highest for sexual IPV (Mean = 3.17 days, SD = 2.42; Table 3).

 

**Table 1. Baseline socio-demographic characteristics of sample (k = 43 clusters, n = 311 individuals).**

| Variable, M(SD) or n(%) | Full sample (n = 311) | Clusters (k = 43) | | Individuals (n = 311) | | | |
|---|---|---|---|---|---|---|---|
| | | Usual care group (k = 21) | Nguvu group (k = 22) | Usual care group (n = 153) | Nguvu group (n = 158) | OR | 95% CI |
| Age (in years), M(SD) | 33.52(9.01) | 33.33(4.82) | 31.40(7.33) | 34.46(8.15) | 32.61(9.70) | 0.98 | .95–1.00 |
| Years living in Nyarugusu, M(SD) | 17.62(4.81) | 17.79(2.75) | 17.81(3.07) | 17.85(4.61) | 17.40(5.00) | 0.98 | .94–1.03 |
| Marital status, cluster mean %(SD), individual n (%) | | | | | | | |
| *Married, living with partner* | 229(73.63) | 73.06(24.77) | 72.38(25.88) | 112(73.20) | 117(74.05) | REF | REF |
| *Married, not living with partner* | 19(6.11) | 9.33(15.43) | 3.66(7.32) | 12(7.84) | 7(4.43) | 0.56 | 0.21–1.47 |
| *In relationship, living with partner* | 21(6.75) | 4.18(8.80) | 8.47(22.74) | 9(5.88) | 12(7.59) | 1.28 | 0.52–3.15 |
| *In relationship, not living with partner* | 42(13.50) | 13.43(16.70) | 15.49(21.13) | 20(13.07) | 22(13.92) | 1.05 | 0.54–2.03 |
| Wabembe ethnicity (ref = other), cluster mean %(SD), individual n(%) | 229(73.63) | 92.23(11.79) | 84.62(22.59) | 112(73.20) | 117(74.05) | 0.50 | 0.24–1.04 |
| Education level, cluster mean %(SD), individual n(%) | | | | | | | |
| *Less than primary/none* | 68(21.86) | 24.70(24.25) | 15.01(18.09) | 41(26.80) | 27(17.09) | REF | REF |
| *Primary school* | 129(41.48) | 38.86(25.77) | 35.43(24.28) | 64(41.83) | 65(41.14) | 1.54 | 0.85–2.80 |
| *Secondary school or higher* | 114 (36.66) | 36.45(30.33) | 49.56(26.71) | 48(31.37) | 66(41.77) | 2.09 | 1.13–3.85 |
| Literate (ref = illiterate), cluster mean %(SD), individual n(%) | 216(73.72) | 74.90(20.73) | 81.50(15.70) | 102(70.83) | 114(76.51) | 1.34 | 0.80–2.26 |
| Religion, cluster mean %(SD), individual n(%) | | | | | | | |
| *Catholic* | 58(18.71) | 18.11(16.00) | 19.95(23.77) | 28(18.30) | 30(19.11) | REF | REF |
| *Methodist/Free/United* | 58(18.71) | 28.09(22.12) | 12.34(17.01) | 39(25.49) | 19(12.10) | 0.45 | 0.21–0.96 |
| *Other Christian religion* | 162(52.26) | 41.78(19.58) | 60.25(21.57) | 70(45.75) | 92(58.60) | 1.23 | 0.67–2.24 |
| *Muslim* | 15(4.84) | 5.59(10.25) | 4.75(8.28) | 7(4.58) | 8(5.10) | 1.07 | 0.34–3.33 |
| *Other* | 17(5.48) | 6.43(15.98) | 2.71(4.76) | 9(5.88) | 8(5.10) | 0.83 | 0.28–2.45 |
| Has children (ref = none), cluster mean %(SD), individual n(%) | 301(96.78) | 98.39(4.20) | 92.98(15.22) | 150(98.04) | 151(95.57) | 0.43 | 0.11–1.70 |
| Number of children, M(SD) | 4.76(2.49) | 4.95(1.29) | 2.04(1.41) | 5.08(2.54) | 4.44(2.41) | 0.90 | 0.82–0.99 |
| Household size, M(SD) | 7.26(3.32) | 7.47(1.76) | 7.38(1.67) | 7.35(3.39) | 7.17(3.27) | 0.98 | 0.92–1.05 |
| Resettlement, cluster mean %(SD), individual n (%) | | | | | | | |
| *Listed on resettlement board* | 113(36.45) | 34.88(21.78) | 35.26(22.01) | 53(34.64) | 60(38.22) | 1.17 | 0.73–1.85 |
| *Interviewed for resettlement* | 103(33.33) | 31.91(22.77) | 30.89(23.55) | 49(32.03) | 54(34.62) | 1.12 | 0.70–1.80 |
| *Completed resettlement health screen* | 15(4.87) | 3.74(12.26) | 6.76(10.59) | 7(4.58) | 8(5.16) | 1.14 | 0.40–3.21 |
| *Participated in cultural orientation sessions* | 10(3.25) | 3.28(12.20) | 2.56(6.56) | 6(3.92) | 4(2.58) | 0.65 | 0.18–2.35 |

(*Continued*)

**Table 1.** (Continued)

| Variable, M(SD) or n(%) | Full sample (n = 311) | Clusters (k = 43) | | Individuals (n = 311) | | OR | 95% CI |
|---|---|---|---|---|---|---|---|
| | | Usual care group (k = 21) | Nguvu group (k = 22) | Usual care group (n = 153) | Nguvu group (n = 158) | | |
| *Would prefer to be resettled* | 307(98.71) | 100.00(0.00) | 98.13(4.71) | 153(100.00) | 154(97.47) | — | — |

Note: In the full sample we report mean (standard deviation) for continuous variables and n (%) for categorical variables. In the cluster columns, we report the cluster mean (standard deviation) for continuous variables as well as the mean proportion (standard deviation) for categorical variables across clusters. For the individual columns we report the mean (standard deviation) for continuous variables and the n (%) for categorical variables.

## Changes in primary and secondary outcomes over time (relevance)

In the full intent-to-treat sample, the reductions in mental health and IPV severity were moderate (Cohen's d = 0.47–0.62 for mental health, d = 0.42 for IPV severity). We identified small changes in functioning (d = 0.15) and IPV frequency (d = 0.04–0.18). Changes in mental health, IPV, and functioning were correlated in the expected directions (i.e., improved mental health associated with decreased IPV and improved functioning). Correlations between change in mental health outcomes over time were strong (r = 0.62–0.92). Changes in the frequency of IPV were strongly correlated for physical and psychological IPV frequency (r = 0.70), and moderately correlated with sexual IPV (psychological r = 0.30, physical r = 0.36). Correlations between change in mental health and IPV outcomes were weak to moderate (r = 0.10–0.30). Functioning was moderately correlated with changes in mental health (r = 0.25–0.44) and weakly correlated with changes in IPV (r = 0.04–0.16).

**Table 2. Summary of findings.**

| Outcome of interest | Variable | Analysis | Result |
|---|---|---|---|
| Relevance | Prevalence of IPV | Proportion of sampling frame and women screened who report past-year IPV | Positive: 85% of women screened reported past-year IPV; At least 50% of women in women's groups (not all were screened) have experienced past-year IPV |
| | Mental health among IPV survivors | Distribution of mental health problems assessed during screening | Positive: 5.7% of women screened reported no or mild symptoms of psychological distress |
| | Sensitivity to change | Effects sizes for outcome change scores; Correlation in change scores for primary and secondary outcomes; Mixed effects regression models comparing differences in outcome measures between study conditions at endline | Mixed: Moderate change in mental health and IPV severity, but small changes in functioning and IPV frequency; Weak to moderate correlations between change in mental health and IPV; Small between-group differences in outcomes after controlling for baseline imbalances |
| Acceptability | Intervention completion | Participated in 6 or more sessions; average attendance | Mixed: more than half (58.2%) completed, average attendance 5.4/8 sessions; higher attendance for one facilitator pair; Intervention completion was lower among women who reported physical IPV, more frequent sexual IPV, or more frequent overall IPV at baseline |
| | Safety | Adverse events | Positive: 3 adverse events identified, none of which were related to study participation |
| Feasibility | Recruitment | Number of eligible women identified | Mixed: Large proportion of women screened were eligible; size of sampling frame limited recruitment |
| | Attrition | % of participants lost to follow-up | Positive: 11.6% lost to follow-up; similar between study conditions |
| | Randomization | Comparison of demographic and wellbeing scores at baseline | Mixed: some imbalances identified, including on primary outcomes |
| | Correct implementation of study procedures | Protocol deviations | Mixed: 1 eligibility screen inaccurately evaluated by research staff and 1 incomplete screening interview; Few protocol deviations |

**Table 3. Baseline clinical characteristics of sample (n = 311).**

| | Full sample (n = 311) | Usual care group (n = 153) | Nguvu group (n = 158) | OR | 95% CI |
|---|---|---|---|---|---|
| MENTAL HEALTH AND FUNCTIONING | M(SD) | M(SD) | M(SD) | | |
| Depressive symptoms (HSCL) | 2.22(0.38) | 2.27(.36) | 2.17(.4) | 0.49 | 0.27–0.90 |
| Anxiety symptoms (HSCL) | 2.17(0.50) | 2.23(.49) | 2.12(.5) | 0.64 | 0.40–1.00 |
| Post-traumatic stress symptoms (HTQ) | 2.24(0.39) | 2.31(.39) | 2.17(.38) | 0.39 | 0.21–0.71 |
| Functional impairment | 1.67(0.75) | 1.68(.8) | 1.67(.70) | 0.99 | 0.73–1.33 |
| INTIMATE PARTNER VIOLENCE | n(%) | n(%) | n(%) | | |
| Controlling behaviors | 298(98.35) | 144(97.96) | 154(98.72) | 1.60 | 0.26–9.74 |
| Any psychological IPV | 209(67.20) | 108(70.59) | 101(63.92) | 0.74 | 0.46–1.19 |
| Any physical IPV | 256(82.32) | 126(82.35) | 130(82.28) | 0.99 | 0.56–1.78 |
| Any sexual IPV | 296(95.18) | 152(99.35) | 144(91.14) | 0.07 | 0.01–0.52 |
| IPV injury in past 2 weeks | 224(98.25) | 116(98.31) | 108(98.18) | 0.93 | 0.13–6.73 |
| | M(SD) | M(SD) | M(SD) | | |
| Past 2-week frequency of psychological IPV | 1.36(1.59) | 1.39(1.55) | 1.34(1.63) | 0.98 | 0.85–1.13 |
| Past 2-week frequency of physical IPV | 0.77(1.16) | 0.86(1.24) | 0.68(1.07) | 0.87 | 0.71–1.06 |
| Past 2-week frequency of sexual IPV | 3.17(2.42) | 3.60(2.5) | 2.75(2.27) | 0.86 | 0.77–0.95 |

The intraclass correlation coefficient across all outcomes was low: depression (ICC = 0.062), functioning (ICC = 0.058), sexual IPV (ICC = 0.032), anxiety (ICC = 0.030), post-traumatic stress (ICC = 0.026), psychological IPV (ICC = 0.007), and physical IPV (ICC<0.001).

## Differences in mental health, IPV, and functional impairment between Nguvu and usual care group at endline (relevance)

In the unadjusted models and those adjusting for unbalanced demographic characteristics (i.e., education, religion, children) we found lower psychological distress for all outcomes (i.e., depression, anxiety, post-traumatic stress) in the Nguvu condition relative to the usual care condition at endline. There was no difference in functional impairment or IPV in the unadjusted models or the models adjusting for baseline demographic differences between Nguvu and the usual care participants at endline. It is notable that psychological IPV, physical IPV, and IPV severity were higher in the Nguvu relative to the usual care condition at endline. After further adjusting for baseline differences in the primary outcomes (depression, anxiety, post-traumatic stress, frequency of sexual IPV), we found an attenuation in the differences between groups on most outcomes. The per protocol analysis that was restricted to Nguvu intervention completers and all usual care participants identified larger between-group differences in mental health and functioning outcomes, including lower levels of post-traumatic stress symptoms in the Nguvu condition relative to the usual care condition at the post-intervention assessment (Mean Diff = -0.22; 95% CI: -0.43, -0.01; Table 4).

Exploratory analyses revealed baseline characteristics that modified the differences between Nguvu and usual care conditions in both psychological distress and IPV frequency. First, we found greater differences in psychological distress favoring the Nguvu condition among people with higher levels of psychological distress, lower levels of IPV frequency, and fewer traumatic events experienced at baseline. We identified higher reported frequencies of IPV at post-intervention among participants with greater psychological distress, IPV frequency, and more traumatic events experienced at baseline (Table 5).

**Table 4. Mean difference in primary and secondary outcomes at post-intervention between Nguvu and usual care group in intent-to-treat and per-protocol analyses.**

| | Intent-to-treat analyses | | | | | | Per-protocol analysis | |
| --- | --- | --- | --- | --- | --- | --- | --- | --- |
| | Model 1: Unadjusted (n = 275) | | Model 2: Adjusting for unbalanced demographics (n = 274) | | Model 3: Adjusting for unbalanced demographics, mental health, and violence at baseline (n = 272) | | Model 4: Adjusting for unbalanced demographics, mental health, and violence at baseline (n = 219) | |
| *Outcomes*: | Mean difference | 95% CI | Mean difference | 95% CI | Mean difference | 95% CI | Mean difference | 95% CI |
| Depression (HSCL) | -0.21 | -0.38, -0.04 | -0.21 | -0.38, -0.04 | -0.14 | -0.30, 0.03 | -0.17 | -0.34, 0.01 |
| Anxiety (HSCL) | -0.22 | -0.40, -0.04 | -0.22 | -0.40, -0.04 | -0.12 | -0.30, 0.06 | -0.12 | -0.32, 0.08 |
| Post-traumatic stress (HTQ) | -0.26 | -0.44, -0.08 | -0.28 | -0.45, -0.10 | -0.18 | -0.36, 0.00 | -0.22 | -0.43, -0.01 |
| Functional impairment | -0.20 | -0.42, 0.02 | -0.21 | -0.43, 0.00 | -0.14 | -0.37, 0.09 | -0.24 | -0.48, 0.01 |
| Psychological IPV | 0.01 | -0.13, 0.16 | 0.01 | -0.13, 0.15 | 0.10 | -0.04, 0.24 | 0.07 | -0.08, 0.24 |
| Physical IPV | 0.03 | -0.09, 0.15 | 0.02 | -0.10, 0.14 | 0.09 | -0.03, 0.21 | 0.05 | -0.08, 0.19 |
| Sexual IPV | -0.08 | -0.28, 0.12 | -0.14 | -0.35, 0.07 | -0.03 | -0.24, 0.17 | 0.04 | -0.21, 0.28 |

Note: Models 1–3 included a random intercept for women's group (randomization cluster).

Psychological/Physical/Sexual IPV frequency outcomes are log-transformed.

Model 2 covariates: Education, religion, number of children.

Model 3 covariates: Depression, anxiety, ptsd, frequency of sexual violence, education, religion, number of children.

## Intervention completion and study attrition (acceptability, feasibility)

Fifteen Nguvu groups that included 6–13 participants each were assembled and implemented throughout the course of the study period. Less than one-third of participants randomized to the Nguvu intervention attended all sessions (n = 47, 29.7%) and 8.2% (n = 13) attended none. On average, Nguvu participants attended 5.4 of 8 sessions. Over half (n = 92; 58.2%) of those assigned to the Nguvu intervention were considered "intervention completers" given that they attended six or more sessions (Fig 1). Attendance was highest in the first session (81.7%) and lowest for the seventh session (57.3%). Most socio-demographic and baseline wellbeing variables were not associated with intervention completion. Exceptions were lower intervention completion by women who reported experiencing physical IPV perpetrated by their current or most recent partner (OR = 0.32, 95% CI: 0.12, 0.84), more frequent past two-week sexual IPV (OR = 0.83, 95% CI: 0.71, 0.96), or more frequent past two-week overall IPV (OR = 0.74, 95% CI: 0.55, 0.99). We also found that participants assigned to one of the five intervention facilitator pairs had greater odds of completing the intervention relative to participants assigned to the other four intervention facilitator pairs (OR = 3.13, 95% CI: 1.41, 6.95).

Research study attrition, defined as not completing the endline interview, was low (n = 36; 11.6%) and did not substantially differ between study conditions (n = 15, 9.5% in Nguvu;

**Table 5. Moderators of the difference in psychological distress and IPV frequency by study condition (n = 275).**

| Outcome: | Psychological Distress | | | | IPV Frequency | | | |
| --- | --- | --- | --- | --- | --- | --- | --- | --- |
| | Moderator low | | Moderator high | | Moderator low | | Moderator high | |
| Moderator: | Mean difference | 95% CI | Mean difference | 95% CI | Mean difference | 95% CI | Mean difference | 95% CI |
| Psychological distress | -0.11 | -0.32, 0.09 | -0.25 | -0.48, -0.02 | -0.02 | -0.47, 0.44 | 0.45 | -0.06, 0.96 |
| IPV Frequency | -0.21 | -0.42, -0.01 | -0.02 | -0.20, 0.16 | -0.27 | -0.69, 0.15 | 0.90 | 0.38, 1.42 |
| Traumatic events (experienced) | -0.24 | -0.49, 0.00 | -0.09 | -0.26, 0.08 | -0.06 | -0.51, 0.38 | 0.46 | -0.03, 0.94 |
| Marital status | -0.25 | -0.57, 0.08 | -0.16 | -0.34, 0.02 | 0.08 | -0.63, 0.78 | 0.16 | -0.25, 0.57 |

n = 21, 13.7% in usual care). The most common reason for attrition was resettlement or relocation outside of the camp (n = 18 out of 36 lost to follow-up). Similarly, the only predictor of attrition was being in the later stages of the resettlement process. More specifically, having had a resettlement health screen or participating in the resettlement orientation process was associated with a 10.8-fold (95% CI: 3.7, 32.1) and 21.6-fold (95% CI: 5.3, 88.3) increase in the odds of study attrition (See S1 File). Refusal to respond to specific assessment items (i.e., item-level missingness) was uncommon. Only one participant withdrew from the study, which was due to concern that her neighbors would find out about her participation.

Three adverse events were identified over the study period by facilitators and research staff. The Data and Safety Monitoring Board and Johns Hopkins Institutional Review Board reviewed these cases and determined that they were unlikely to be due to study participation and consistent with ongoing IPV that the participant had been experiencing. In all cases of potentially adverse events, study staff followed up with participants to evaluate their safety and implemented appropriate procedures to connect them with health and protection services.

## Discussion

In this study we found Nguvu to be relevant to the needs of Congolese refugee women in Nyarugusu affected by IPV and psychological distress (see Table 2 for a summary of main findings). A large proportion of the women in the sampling frame, refugee women participating in local women's groups, were eligible for the Nguvu intervention suggesting a high burden of IPV and psychological distress in this community. Furthermore, IPV was severe and most women reported having had experienced IPV-related injuries at baseline. While we found evidence of a significant burden of psychological distress and IPV in these communities, we found mixed evidence regarding the change in these constructs over time and the differences in mental health and IPV by study condition post-intervention. It appeared that mental health was more sensitive to change, but between-group differences revealed small effect sizes that may not be clinically significant.

With regard to acceptability, we did not identify any adverse events that were determined to be related to participation in this study. Despite efforts to improve intervention retention and completion, we were unable to increase the average number of sessions attended by participants compared to the previous intervention cohort study [23]. Similar to previous studies of psychological interventions for IPV survivors [46], we found lower rates of intervention and session attendance among women who had experiencing physical IPV and more frequent sexual or overall IPV suggesting that the Nguvu intervention may be less acceptable and feasible for women with more severe IPV profiles. Various factors can impact attendance for these women. Women who are currently in violent relationships are *ipso facto* exposed to ongoing threat and actual violence, which can impact how they can practically engage with the outside world, including the decision to attend intervention sessions. Women in current violent relationships must make decisions regarding when and how they engage with others, or leave the home to attend events, based on the risk to them or their children as a result of that judgement. This is particularly true in crowded refugee camp settings when privacy and space are limited. Interventions that can be flexibly attended, accessible, safe, and appropriate for women experiencing severe IPV are needed [47].

A primary objective of this trial was to evaluate the feasibility of implementing and evaluating the integrated Nguvu intervention in a refugee setting. To enhance feasibility this study implemented a shortened version of Cognitive Processing Therapy and Advocacy Counseling relative to the previous trials that evaluated these respective interventions independently in populations affected by gender-based violence [31,40]. Relative to the original Cognitive Processing Therapy trial in the eastern DRC we found smaller effect sizes for mental health

outcomes [31]. These differences may be attributable to the changes made to the intervention (e.g., shortening of intervention, integrated IPV and mental health intervention), differences between the study populations (e.g., displaced population with restrictions on work/movement and little hope for resettlement, differences in burden of psychological distress and IPV), or implementation factors (e.g., less intensive supervision, multi-sectoral intervention, availability of complementary IPV and mental health services). Although preliminary and underpowered, we generally found that the intervention showed better results for mental health outcomes, particularly among those with higher levels of psychological distress and lower levels of IPV. Similar to previous trials of advocacy counseling, our results on the IPV changes and effect sizes were mixed and inconclusive [40]. It is likely that the Nguvu intervention would require a more detailed IPV component to reduce violence, particularly for women with more severe IPV, that is relevant to the complex refugee context.

While we were able to train lay refugee incentive workers to deliver the intervention, we identified differences in the likelihood of participant intervention completion by facilitator pair suggesting possible heterogeneity in the quality of implementation across facilitators. Future studies may consider strategies for improving training and implementation to increase fidelity to the intervention and research protocols; however, it is important to consider how upgrading training and supervision may compromise the real-world validity of findings in complex, low-resource humanitarian settings, such as Nyarugusu refugee camp.

The current study possessed several limitations that should be considered when interpreting these results and planning future research on integrated mental health and IPV interventions in refugee settings. As indicated in our study protocol [32], we intended to enroll 400 refugee women affected by IPV with elevated psychological distress in Nyarugusu. After exhausting our a priori sampling frame and recruitment procedures we enrolled 311 women. Therefore, our analyses were not powered to detect significant between-group differences in outcomes at the endline assessment. Similarly, this study was not powered to detect subgroup differences and thus moderation analyses are considered exploratory. Despite being underpowered, there remains an elevated risk of Type I error due to multiple testing in the moderation analyses. Despite efforts to improve the reliability of the self-reported measurement instruments, particularly poor test-retest reliability of the sexual violence measure and moderate internal consistency of the functional impairment measure, residual measurement issues may have contributed imprecision to our estimates. It is also possible that the lack of masking of study participants may have influenced reporting. Furthermore, while the research assistants were not informed of the participants' allocation, it is plausible that information divulged during the assessments may indicate whether they were participating in the Nguvu intervention, of which the research assistants were familiar and may have introduced biases in outcome assessment.

Importantly, the internal validity of the sensitivity to change and moderation findings may have been compromised by baseline imbalances between study conditions and differential intervention completion by IPV severity. While we did find small improvements in the mental health outcomes, most of these findings were confounded by between-group differences in demographics, mental health, and IPV at baseline. It is possible that there are other unobserved confounders that were not measured in this study and may also explain some of the between-group differences and observed change in the primary outcomes over time. Our inferences are also complicated by lower rates of intervention completion among participants reporting any physical IPV and more frequent sexual and overall IPV at baseline. The moderator analyses similarly suggest that Nguvu was associated with better outcomes for women with less severe IPV profiles, which may, in part, be explained by the lower dose of intervention received by women with more severe IPV who were more likely to drop out, possibly because they did not find it useful or safe. Although we had reasonable rates of follow-up in the

research interviews, we found that participants in the later stages of the resettlement process were more likely to drop out because they had been relocated outside of the camp by the time of their endline interview. This non-random process may have introduced bias into our longitudinal findings because there could have been relevant differences between participants by stage of resettlement that may impact their change in mental health or IPV over time as well as their response to the Nguvu intervention. However, the degree of attrition due to resettlement was similar between groups. Therefore, to preserve the internal validity of a future definitive trial in displaced populations seeking resettlement or repatriation, it will be critical to account for potential study attrition in the design and analysis.

Despite these limitations, this study is the first to evaluate the feasibility of an integrated, multi-sectoral IPV and mental health intervention delivered by lay providers in a refugee setting. Using the data generated from this feasibility trial we identified factors related to intervention retention and response, including potential subgroups that may be more (or less) responsive to the intervention, as well as important implementation considerations. Findings from this randomized feasibility trial also provide several critical recommendations for future definitive trials intended to evaluate the effectiveness and/or implementation of an integrated IPV and mental health intervention, such as Nguvu, in humanitarian settings. First, randomizing more clusters and/or using stratified or other randomization procedures to eliminate between-group baseline imbalances is needed to preserve the internal validity of future cluster-randomized controlled trials of Nguvu [25]. An alternative approach is to consider individual randomization. In this study context, cluster randomization was preferred as women's groups provided a safe setting for recruitment and screening, women preferred to attend groups along with their peers, and there was a high risk of contamination within women's groups [23]. Second, it is imperative that future definitive trials improve retention, which may require modified intervention retention strategies, flexible scheduling and accommodations, more dedicated and detailed training, or other approaches. Results from a forthcoming process evaluation conducted upon completion of this feasibility trial may provide insight into strategies for improving retention and factors that contributed to intervention non-completion as well as other implementation challenges and considerations. Third, trial eligibility should be closely examined to identify a range of severity in IPV and psychological distress that is acceptable and appropriate to the level of intervention provided. Overall, results from this feasibility trial indicate that an integrated approach to addressing IPV and mental health is relevant to the needs of refugee women; however, the implementation strategies employed to promote feasibility (e.g., shortened intervention, task shifting, and realistic levels of training and supervision) may make this approach most acceptable for women with less severe IPV histories.

## Supporting information

**S1 File. Correlates of intervention completion and study attrition.** Model results describing participant characteristics that were associated with intervention completion and study attrition.
(DOCX)

**S2 File. Johns Hopkins Institutional Review Board protocol.** Study protocol approved by the Johns Hopkins Institutional Review Board.
(DOCX)

**S3 File. CONSORT checklist.** CONSORT checklist for reporting of feasibility trials.
(DOC)

## Acknowledgments

We would like to thank UNHCR and IRC Kasulu field office staff for their assistance in field operations. We are very grateful to the refugee incentive workers who served as research assistants and intervention facilitators, without whom this research would not have been possible. We would also like to thank the women who participated in this study for generously sharing their time and experiences with us.

## Author Contributions

**Conceptualization:** Susan Rees, Peter Ventevogel, Wietse A. Tol.

**Data curation:** Wietse A. Tol.

**Formal analysis:** M. Claire Greene.

**Funding acquisition:** Wietse A. Tol.

**Investigation:** M. Claire Greene, Annie Bonz, Debra Kaysen, Peter Ventevogel, Wietse A. Tol.

**Methodology:** M. Claire Greene, Samuel Likindikoki, Debra Kaysen, Rachael Turner, Peter Ventevogel, Jessie K. K. Mbwambo, Wietse A. Tol.

**Project administration:** M. Claire Greene, Samuel Likindikoki, Annie Bonz, Lusia Misinzo, Tasiana Njau, Shangwe Kiluwa, Rachael Turner, Wietse A. Tol.

**Resources:** Wietse A. Tol.

**Supervision:** M. Claire Greene, Samuel Likindikoki, Lusia Misinzo, Tasiana Njau, Shangwe Kiluwa, Rachael Turner, Jessie K. K. Mbwambo, Wietse A. Tol.

**Writing – original draft:** M. Claire Greene, Wietse A. Tol.

**Writing – review & editing:** M. Claire Greene, Samuel Likindikoki, Susan Rees, Annie Bonz, Debra Kaysen, Lusia Misinzo, Tasiana Njau, Shangwe Kiluwa, Rachael Turner, Peter Ventevogel, Jessie K. K. Mbwambo, Wietse A. Tol.

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
