## [Decision Letter · Decision Letter 0]

29 Jul 2020

PONE-D-20-13577

Evaluation of an integrated intervention to reduce psychological distress and intimate partner violence in refugees: Results from the Nguvu cluster randomized feasibility trial

PLOS ONE

Dear Dr. Greene,

Thank you for submitting your manuscript to PLOS ONE. After careful consideration, we feel that it has merit but does not fully meet PLOS ONE’s publication criteria as it currently stands. Therefore, we invite you to submit a revised version of the manuscript that addresses the points raised during the review process.

We look forward to receiving your revised manuscript.

Kind regards,

Nancy Beam, PhD

Academic Editor

PLOS ONE

Journal Requirements:

Reviewers' comments:

Reviewer's Responses to Questions

**Comments to the Author**

1. Is the manuscript technically sound, and do the data support the conclusions?

Reviewer #1: Yes

Reviewer #2: Partly

2. Has the statistical analysis been performed appropriately and rigorously? 

Reviewer #1: Yes

Reviewer #2: Yes

3. Have the authors made all data underlying the findings in their manuscript fully available?

Reviewer #1: Yes

Reviewer #2: No

4. Is the manuscript presented in an intelligible fashion and written in standard English?

Reviewer #1: Yes

Reviewer #2: Yes

5. Review Comments to the Author

Reviewer #1: Congratulations to the authors for tackling a complex and difficult topic in an extremely challenging environment. It is a Herculean task but one worth doing. The study is the result of careful published developmental work and was carefully implemented in the very constraining context. This reviewer appreciated the clarity and honesty of the discussion section. I think the lessons from the study are important for others seeking to adapt the methods to their own studies. However, I think that more needs to be explained and the lessons made clearer.

The authors have addressed several of the major limitations of this study in their discussion, but I have concerns that I think need addressing before the article should be published.

1. The study is described as a feasibility cluster randomised trial, but it is not sufficiently well described. It is as if it is individually randomised.

a. Please provide a table describing the characteristics of the clusters. (You yourselves have described the differentiation between trainers inside those clusters).

b. The flow chart should begin with eligible clusters, followed by women. Please add to your flowchart detail of how many women were randomised into each group, following the table with cluster characteristics and then detail of any cluster dropouts as well as those individual women in each cluster

2. Pls include a summary of the power calculation, effect size sought and ICC in your methods to reach n=400 outlined in your protocol prior to statistical analyses.

3. Please alter references for the CTS. CTS 1 has been widely criticised. ORC Macro is a website for an American development aid organisation that requires login or registration to access. Please provide a more accessible reference or cite CTS2 if that is the adaptation.

4. P.8 (under population) you add screening questions about phys/sex violence (probably better in procedures /recruitment section. P.11 you state measurement of psych/phys & sexual violence. How was psychological violence measured?

5. Statistical analyses. What do you mean by intent-to-treat? Do you mean inclusion of the 275 completed cases by arm?

6. You did very well with maintaining the sample 88%, but you did have one in ten women dropout. Even though they were (as always) the more severe, how was missingness accounted for? Did you use any imputation?

7. How was missingness taken into account for composite variables?

8. Minor points:

a. Please reference your measures in the first section as well as later on.

b. P.6. line98 – Please explain or give a reference for ‘qualitative free listing’ also for a ‘refugee incentive worker’ for those unfamiliar with these settings

c. Please add numbers of your ethics review boards

d. Please add n= at the top of tables five and table six.

e. Where is Table 4? You may need to relabel

f. P.20 onwards , please give number/denominator and %s.

Reviewer #2: This paper presents a “feasibility” seeking cluster-randomized trial (CRT). As per the convention, no sample size justification is provided. Albeit, the paper fall short on several areas which I will focus bellow.

1. Description of randomization is either incomplete or missing. This is very important as the randomization in CRT is much more challenging. Please describe clearly randomization scheme. A feasibility aspect of randomization needs to be discussed.

2. Why CRT is chosen as compared to simple RCT also need to be discussed. For example what consist of a cluster unit and why individual randomization to one of the arm cannot be done should be discussed.

3. Report Intra-Class-Correlation as that is useful for a future powered trial.

4. Following the notion of feasibility trial all results of significance and p-value must be removed. Note, a pilot or feasibility trial is not efficacy establishing trial and as a result, no claim of significance can be made. Confidence intervals are fine as well as text indicating the direction of effect size.

5. While no Sample size justification is not necessary I wonder how authors came up with n=311!! How many clusters per arm is there? Note, this is pretty large sample size for a feasibility study which typically has much restricted sample size. This issue needs clarification.

6. It is not clear whether authors encountered any missing data and how it is handled or plans to be handled in the powered future trial.

Also explain how the data will be made available as per PLOS-policy.

6. PLOS authors have the option to publish the peer review history of their article (what does this mean?). If published, this will include your full peer review and any attached files.

Reviewer #1: No

Reviewer #2: No

---

## [Author Response · Author response to Decision Letter 0]

4 Sep 2020

Reviewer Comments

R1.1 Congratulations to the authors for tackling a complex and difficult topic in an extremely challenging environment. It is a Herculean task but one worth doing. The study is the result of careful published developmental work and was carefully implemented in the very constraining context. This reviewer appreciated the clarity and honesty of the discussion section. I think the lessons from the study are important for others seeking to adapt the methods to their own studies. However, I think that more needs to be explained and the lessons made clearer. The authors have addressed several of the major limitations of the study in their discussion, but I have concerns that I think need addressing before the article should be published.

Response: Thank you for your reflections on the paper. We have noted the important suggestions you have made and detailed our responses to each of your comments below.

R1.2 The study is described as a feasibility cluster randomized trial, but it is not sufficiently well described. It is as if it is individually randomized.

Response: We have rearranged the ‘Procedures’ section and added additional text to clarify the cluster randomized design of the feasibility trial. Furthermore, we have revised the flow diagram as recommended in comment R1.4 to show the randomization of clusters (i.e., women’s groups) followed by the recruitment of individual women within those clusters. We hope this is now articulated more clearly in the paper.

“We employed a cluster randomized feasibility trial design in order to build upon the existing infrastructure of local women’s groups (i.e., clusters), which are organized at the village-level by the International Rescue Committee (20). These women’s groups are aimed at strengthening social networks and providing skills training. Women’s groups focused on a variety of skills and objectives including cooking, village savings and loans, weaving, etc. Women’s groups were randomized to the Nguvu intervention versus usual care condition using a random number generator in Stata, Version 14 with 1:1 allocation. 

We obtained a list of women’s groups from the International Rescue Committee and found that 43 of the 63 listed women’s groups were operational. In these 43 women’s groups there were 647 members. We approached the leaders of the 43 women’s groups to receive their permission to deliver a brief summary of the study and invite interested members to participate. The study was presented to the women’s group members as a study of women’s health and wellbeing to avoid interested women being identified by their peers as IPV survivors. All 43 operational clusters were randomized and agreed to voluntary recruitment, screening and enrollment of their members. Women recruited for the study were allocated to study condition based on the randomization of women’s groups (i.e., clusters). Women recruited from women’s groups randomized to Nguvu were enrolled in the intervention, whereas none of the women who were recruited from women’s groups randomized to the control received the intervention.” Pgs. 9-10

R1.3 Please provide a table describing the characteristics of the clusters. (You yourselves have described the differentiation between the trainers inside those clusters).

Response: We have added characteristics of clusters to table 1 and described observed differences in the results section.

“At baseline participants were 33.5 (SD=9.0) years of age and had lived in Nyarugusu for 17.6 (SD=4.8) years on average. Most participants were married and living with their partner (73.6%), Wabembe ethnicity (73.6%), literate (73.7%), and Christian (18.7% Catholic, 18.7% Methodist/Free/United, 52.3% other Christian denomination). About one-third had a secondary school education or higher. Almost all participants had children (96.7%; 4.8 children, on average). The average household size was 7.3 (SD=3.3) people. Almost all participants preferred to be resettled, but approximately one-third had begun the process. Few reported completing later stages of the resettlement process (e.g., 4.9% completed a resettlement health screen, 3.3% had participated in cultural orientation sessions). At the cluster level, the distribution of these characteristics differed across conditions. The proportion of women who were not married, but living with their partner, more highly educated, and literate appeared to be higher in women’s groups allocated to the Nguvu condition. The mean number of children appeared higher in clusters allocated to usual care. At the individual level, the distribution of these demographic characteristics was similar across participants allocated to the Nguvu and usual care conditions with few exceptions. Participants in the Nguvu condition were more likely to have a secondary education, less likely to report their religious affiliation as Methodist/Free/United, and had fewer children, on average (Table 1).” Pg. 17

R1.4 The flow chart should begin with eligible clusters, followed by women. Please add to your flowchart detail of how many women were randomized into each group, following the table with cluster characteristics and then detail of any cluster dropouts as well as those individual women in each cluster.

Response: We have updated the flow chart as per your suggestions so the cluster randomization precedes the description of screening and allocation of individuals. In the first paragraph of the results we describe the number of clusters that were randomized. While we did not have any clusters who refused to participate or dropped out, there was one cluster that did not have any eligible women among those screened, which we describe in the first paragraph of the results section. We also added that all clusters agreed to participate and were randomized to the methods section.

“With the exception of one of the women’s groups assigned to the Nguvu condition, at least one woman from all groups was enrolled in the study (median number of women enrolled per women’s group=6; IQR=3,10).” Pg. 16

“We obtained a list of women’s groups from the International Rescue Committee and found that 43 of the 63 listed women’s groups were operational. In these 43 women’s groups there were 647 members. We approached the leaders of the 43 women’s groups to receive their permission to deliver a brief summary of the study and invite interested members to participate. The study was presented to the women’s group members as a study of women’s health and wellbeing to avoid interested women being identified by their peers as IPV survivors. All 43 operational clusters were randomized and agreed to voluntary recruitment, screening and enrollment of their members.” Pg. 9

R1.5 Please include a summary of the power calculation, effect size sought, and ICC in your methods to reach n=400 outlined in your protocol prior to statistical analyses

Response: We have added more details about the power calculation included in the original protocol to the methods section.

“We planned to enroll 400 participants in this study, which would allow us to identify a small to moderate effect size. Enrolling 200 participants from the 63 original women’s groups with approximately equal allocation to each study condition was necessary to achieve at least 80% power to detect a small to moderate effect of the intervention on mental health outcomes accounting for up to 20% attrition and an intraclass correlation ranging from 0.1-0.5 within subjects and 0.1-0.3 within cluster.” Pg. 10

R1.6 Please alter your references for the CTS. CTS 1 has been widely criticized. ORC Macro is a website for an American development aid organization that requires login or registration access. Please provide a more accessible reference or cite CTS2 if that is the adaptation.

Response: We have updated the references so the original survey is accessible and have clarified that we used the CTS adaptation developed for the Demographic and Health Surveys.

“Psychological, physical, and sexual IPV were measured in the baseline and follow-up assessment using an adapted version of the Conflict Tactics Scales that was developed for the World Health Organization multi-country study on women’s health and domestic violence against women and the Demographic and Health Surveys (25-27).” Pg. 12

R1.7 P.8 (under population) you add screening questions about phys/sex violence (probably better in procedures/recruitment section. P.11 you state measurement of psych/phys & sexual violence. How was psychological violence measured.

Response: We moved the description of screening and eligibility to the procedures section. At screening, we used the brief Abuse Assessment Screen to measure past-year physical or sexual violence, which was an inclusion criterion. Eligible, enrolled women completed a more in-depth assessment of psychological, physical, and sexual violence using the adapted version of the Conflict Tactics Scales developed for the Demographic and Health Surveys and the WHO multi-country study. We have added details about these measures to the methods section to clarify the differences in measurement at screening as compared to baseline and follow-up assessments. We have included a reference to the original measure for the baseline/follow-up assessment, which will allow readers to see the original physical, psychological, and sexual violence questions.

“At screening, physical and sexual violence were assessed using the Abuse Assessment Screen (24). Psychological, physical, and sexual IPV were measured in the baseline and follow-up assessment using an adapted version of the Conflict Tactics Scales that was developed for the World Health Organization multi-country study on women’s health and domestic violence against women and the Demographic and Health Surveys (25-27). IPV frequency was calculated as the mean reported frequency of each type of IPV (physical, psychological, sexual) over the past two-weeks.” Pg. 12

R1.8 Statistical analyses. What do you mean by intent-to-treat? Do you mean inclusion of the 275 completed cases by arm?

Response: By intent-to-treat, we mean ‘analyzed as randomized’ whereby participants were grouped into study conditions based on randomization as opposed to what intervention they received. This was a complete case analysis as well. We have added this information to the statistical analyses section.

“To examine whether we were able to detect between-group differences we estimated the difference in means between the Nguvu and usual care condition at the post-intervention assessment. These intention-to-treat, complete case analyses included participants who completed the follow-up assessment and assigned study condition membership according to randomization. Models included random intercepts for women’s group to account for clustering.” Pg. 14

R1.9 You did very well with maintaining the sample 88%, but you did have one in ten women dropout. Even though they were (as always) the more severe, how was missingness accounted for? Did you use any imputation?

Response: For this feasibility trial we were not aiming to estimate the effect of the intervention. Therefore, we did not think it necessary to impute outcome data for these participants. We explored predictions of retention in treatment and attrition from the research study as important indicators of acceptability and feasibility, respectively, and have reported these results in the paper. We agree that for a definitive trial it will be important that attrition is minimized to preserve the internal validity of the study and have mentioned this in our discussion section.

“Although we had reasonable rates of follow-up in the research interviews, we found that participants in the later stages of the resettlement process were more likely to drop out because they had been relocated outside of the camp by the time of their endline interview. This non-random process may have introduced bias into our longitudinal findings because there could have been relevant differences between participants by stage of resettlement that may impact their change in mental health or IPV over time as well as their response to the Nguvu intervention. However, the degree of attrition due to resettlement was similar between groups. Therefore, to preserve the internal validity of a future definitive trial in displaced populations seeking resettlement or repatriation, it will be critical to account for potential study attrition in the design and analysis.” Pgs. 31-32

R1.10 How was missingness taken into account for composite variables?

Response: Most of the missing data resulted from not completing the follow-up assessment. We had very little item-level missingness and composite variables were relatively complete. In the fully adjusted models reported in Table 5, there were only 3 additional participants excluded from these models due to item-level missingness. We added a sentence to the results mentioning this alongside the research acceptability and attrition finding as well as the sample sizes for each of the models in Table 5 to indicate missingness in the outcome models.

“Research study attrition, defined as not completing the endline interview, was low (11.6%) and was not significantly different between study conditions (9.5% in Nguvu, 13.7% in usual care; p=0.246). The most common reason for attrition was resettlement or relocation outside of the camp (n=18 out of 36 lost to follow-up). Similarly, the only predictor of attrition was being in the later stages of the resettlement process. More specifically, having had a resettlement health screen or participating in the resettlement orientation process was associated with a 10.8-fold (95% CI: 3.7, 32.1) and 21.6-fold (95% CI: 5.3, 88.3) increase in the odds of study attrition. Refusal to respond to specific assessment items (i.e., item-level missingness) was uncommon. Only one participant withdrew from the study, which was due to concern that her neighbors would find out about her participation.” Pgs. 22-23

R1.11 Please reference your measures in the first section as well as later on

Response: Thank you for noting this. We have added in-text citations to accompany all references to the measures in the manuscript. 

R1.12 P.6 line 98 – Please explain or give a reference for ‘qualitative free listing’ also for ‘refugee incentive worker’ for those unfamiliar with these settings

Response: We added a reference for the qualitative methods and definition for ‘refugee incentive worker’ in the introduction.

“To contextualize and strengthen the relevance of this intervention within a protracted refugee setting, we conducted qualitative free listing and key informant interviews with Congolese refugee incentive workers living in Nyarugusu Refugee Camp, which is located in northwestern Tanzania (14). Incentive workers are refugees who undertake work related to the provision of humanitarian assistance and receive fixed compensation referred to as an ‘incentive’ (15).” Pg. 6

R1.13 please add numbers of your ethics review boards

Response: We added IRB numbers for the 3 ethical review committees who approved the protocol.

“All procedures were reviewed and approved by the Johns Hopkins Bloomberg School of Public Health Institutional Review Board (IRB0007219), the Muhimbili University of Health and Allied Sciences Institutional Review Board (2014-10-27/AEC/Vol.X/56), and the Tanzania National Institute for Medical Research (NIMR/HQ/R.8a/Vol.IX/2016).” Pg. 11

R1.14 Please add n= at the top of tables five and table six

Response: We added the sample sizes for each of the models in table 5 and the overall analysis in table 6.

R1.15 Where is table 4? You may need to relabel

Response: Thank you for catching this error. We have renumbered the tables appropriately.

R1.16 P.20 onwards, please give number/denominator and %s

Response: We now report both frequencies and percentages in the text when describing study results. 

R2.1 The paper presents a ‘feasibility’ seeking cluster-randomized trial (CRT). As per the convention, no sample size justification is provided. Albeit, the paper falls short on several areas which I will focus below.

Response: Thank you for your review and helpful comments on the paper. As per reviewer 1’s suggestion, we have added more details about the originally planned sample size calculation, but acknowledge that this study did not achieve this sample size or level of power. Please see below our detailed responses to your suggestions.

R2.2 Description of randomization is either incomplete or missing. This is very important as the randomization in CRT is much more challenging. Please describe clearly randomization scheme. A feasibility aspect of randomization needs to be discussed.

Response: We have added detail to our description of the randomization and allocation process in the procedures section along with a reference providing further information about different randomization approaches, including unrestricted simple randomization, for cluster randomized trials. In the discussion we note that this randomization approach was insufficient and did not result in exchangeability. Future definitive trials should consider another randomization approach (e.g., stratified randomization) to overcome the limitations described in this paper.

“Women’s groups were allocated to the Nguvu intervention versus usual care condition using a simple, unrestricted randomization approach with approximately equal allocation. The random number sequence was generated in Stata, Version 14.” Pgs. 8-9

R2.3 Why CRT is chosen as compared to simple RCT also needs to be discussed. For example what consist of a cluster unit and why individual randomization to one of the arms cannot be done should be discussed.

Response: We have provided a rationale for cluster as opposed to individual randomization in the methods and the discussion. In this study context women’s groups served as a safe location for recruitment and screening and women preferred to participate in groups along with their peers. To avoid contamination and to leverage the existing peer support within these women’s groups, we employed a cluster randomized design for this feasibility trial. We note the limitations of this approach, particularly covariate imbalances, in our results and discussion sections.

“We employed a cluster randomized feasibility trial design in order to build upon the existing infrastructure of local women’s groups (i.e., clusters), which are organized at the village-level by the International Rescue Committee (22). These women’s groups are aimed at strengthening social networks and providing skills training. Women’s groups focused on a variety of skills and objectives including cooking, village savings and loans, weaving, etc. Given the strength of these networks and the potential for contamination within women’s groups, we elected to randomize women’s groups as clusters as opposed to randomizing eligible women individually. Women’s groups were allocated to the Nguvu intervention versus usual care condition using a simple, unrestricted randomization approach with approximately equal allocation (24). The random number sequence was generated in Stata, Version 14.“ Pg. 8-9

“First, randomizing more clusters and/or using stratified or other randomization procedures to eliminate between-group baseline imbalances is needed to preserve the internal validity of future cluster-randomized controlled trials of Nguvu (24). An alternative approach is to consider individual randomization. In this study context, cluster randomization was preferred as women’s groups provided a safe setting for recruitment and screening, women preferred to attend groups along with their peers, and there was a high risk for contamination within women’s groups (22).” Pg. 33 

R2.4 Report intra-class-correlation as that is useful for a future powered trial.

Response: Thank you for this suggestion. We have added the ICCs for each of the outcomes to the results section.

“The intraclass correlation coefficient across all outcomes was low: depression (ICC=0.062), functioning (ICC=0.058), sexual IPV (ICC=0.032), anxiety (ICC=0.030), post-traumatic stress (ICC=0.026), psychological IPV (ICC=0.007), and physical IPV (ICC<0.001).” Pg. 24

R2.5 Following the notion of feasibility trial all results of significance and p-value must be removed. Not, a pilot or feasibility trial is not efficacy establishing trials and as a result, no claim of significance can be made. Confidence intervals are fine as well as text indicating the direction of effect size.

Response: Thank you for this reminder to avoid overstating the findings from this feasibility trial. We have removed p-values and references to ‘significant’/’non-significant’ results from the paper and tables. 

R2.6 While no sample size justification is not necessary I wonder how authors came up with n=311! How many clusters per arm is there? Not, this is pretty large sample size for a feasibility study which typically has much restricted sample size. This issue needs clarification.

Response: We have revised our flow chart to include the number of clusters randomized to each study condition (see Figure 1). We also added explanation of the sample size to the methods section (Pg. 10; R1.5 comment). 

R2.7 It is not clear whether authors encountered any missing data and how it is handled or plans to be handled in the powered future trial

Response: Thank you for noting that we did not describe missing data nor how it was handled. As noted in our reply to reviewer 1’s comments (R1.9; R1.10), most of our missing data was due to research study attrition (approximately 11% of our sample) as opposed to item-level missingness. We have added the sample size for all models in Table 4, which shows that among participants who completed the follow-up interview, only 3 participants were dropped from the fully adjusted models due to item-level missingness (see R1.10). In our discussion we elaborated on study attrition as a limitation and recommendations for addressing drop out (see R1.9). In this feasibility trial we examined correlates of attrition and found that resettlement was the primary reason and predictor of attrition, which was of comparable magnitude in both groups. Given that the objective of this feasibility trial was to examine missingness as an indication of acceptability of procedures and the feasibility of a definitive trial (not to estimate intervention effects), we did not impute missing data points or employ other missing data procedures; however, we do recommend in our discussion that this be considered for a full trial.

R2.8 Also explain how the data will be made available as per PLOS-policy.

Response: Our local IRB protocols specify that only the research team will have access to the data. This level of restriction is necessary given the sensitive nature of the data, the vulnerability of the population (i.e., female refugees experiencing intimate partner violence who are members of local women’s groups), and the potential identifiability of participants within Nyarugusu refugee camp (n=311 enrolled women out of our sampling frame of 647 women who are part of IRC’s women’s groups). Identification of participant who have reported intimate partner violence has the potential to put women at significant risk. We have provided the email address for the Research Data Management contact at the University of Copenhagen, who will review all data requests and will require IRB approval from the individual’s home institution in order to conduct secondary analysis of these data in effort to protect the safety of research participants.

Editorial Comments

E.1 Please ensure that your manuscript meets PLOS ONE’s style requirements, including those for file naming. The PLOS ONE style templates can be found at https://journals.plos.org/plosone/s/file?id=wjVg/PLOSOne_formatting_sample_main_body.pdf and

Response: Thank you for providing these helpful references. We have modified the formatting of the paper to comply with PLOS ONE’s style requirements.

E.2 We note that you have indicated that data from this study are available upon request. PLOS only allows data to be available upon request if there are legal or ethical restrictions on sharing data publicly. For information on unacceptable data access restrictions, please see http://journals.plos.org/plosone/s/data-availability#loc-unacceptable-data-access-restrictions

Response: Our local IRB protocols specify that only the research team will have access to the data. This level of restriction is necessary given the sensitive nature of the data, the vulnerability of the population (i.e., female refugees experiencing intimate partner violence who are members of local women’s groups), and the potential identifiability of participants within Nyarugusu refugee camp (n=311 enrolled women out of our sampling frame of 647 women who are part of IRC’s women’s groups). Identification of participant who have reported intimate partner violence has the potential to put women at significant risk. We have provided the email address for the Research Data Management contact at the University of Copenhagen, who will review all data requests and will require IRB approval from the individual’s home institution in order to conduct secondary analysis of these data in effort to protect the safety of research participants.

E.3 In your revised cover letter please address the following prompts: a) If there are ethical or legal restrictions on sharing a de-identified data set, please explain them in detail (e.g., data contain potentially identifying or sensitive patient information) and who has imposed them (e.g., an ethics committee). Please also provide contact information for a data access committee, ethics committee, or other institutional body to which data requests may be sent; b) If there are not restrictions, please upload the minimal anonymized data set necessary to replicate your study findings as either Supporting Information files or to a stable, public repository and provide us with the relevant URLs, DOIs, or accession numbers. Please see http://www.bmj.com/content/340/bmj.c181.long for guidelines on how to de-identify and prepare clinical data for publication. For a list of acceptable repositories, please see http://journals.plos.org/plosone/s/data-availability#loc-recommended-repositories. We will update your Data Availability statement on your behalf to reflect the information you provide.

Response: Thank you for offering to update our data availability statement. We have responded to the two issues you addressed in the cover letter. 

E.4 Please include captions for your Supporting Information files at the end of your manuscript, and update any in-text citations to match accordingly. Please see our Supporting Information guidelines for more information: http://journals.plos.org/plosone/s/supporting-information

Response: We have renamed our supporting information files and added captions at the end of our manuscript.

---

## [Decision Letter · Decision Letter 1]

8 Mar 2021

PONE-D-20-13577R1

Evaluation of an integrated intervention to reduce psychological distress and intimate partner violence in refugees: Results from the Nguvu cluster randomized feasibility trial

PLOS ONE

Dear Dr. Greene,

Thank you for submitting your manuscript to PLOS ONE. After careful consideration, we feel that it has merit but does not fully meet PLOS ONE’s publication criteria as it currently stands. Therefore, we invite you to submit a revised version of the manuscript that addresses the points raised during the review process.

Thank you for this resubmission. I apologize about the delay in delivering feedback as it was necessary to assign a new academic editor. The reviewers note that many of their original comments and suggestions were attended to with Revision #1; however, they still note that some minor revisions are necessary before publication. We look forward to receiving this second revision. 

We look forward to receiving your revised manuscript.

Kind regards,

Michelle L. Munro-Kramer, PhD, CNM, FNP-BC

Academic Editor

PLOS ONE

Journal Requirements:

Reviewers' comments:

Reviewer's Responses to Questions

**Comments to the Author**

1. If the authors have adequately addressed your comments raised in a previous round of review and you feel that this manuscript is now acceptable for publication, you may indicate that here to bypass the “Comments to the Author” section, enter your conflict of interest statement in the “Confidential to Editor” section, and submit your "Accept" recommendation.

Reviewer #1: (No Response)

Reviewer #3: (No Response)

Reviewer #4: (No Response)

2. Is the manuscript technically sound, and do the data support the conclusions?

Reviewer #1: Yes

Reviewer #3: Partly

Reviewer #4: Yes

3. Has the statistical analysis been performed appropriately and rigorously? 

Reviewer #1: I Don't Know

Reviewer #3: No

Reviewer #4: Yes

4. Have the authors made all data underlying the findings in their manuscript fully available?

Reviewer #1: No

Reviewer #3: Yes

Reviewer #4: Yes

5. Is the manuscript presented in an intelligible fashion and written in standard English?

Reviewer #1: Yes

Reviewer #3: Yes

Reviewer #4: Yes

6. Review Comments to the Author

Reviewer #1: 1.2 Thank you for revising the flowchart table and recruitment description. Your description of the design of the study is much clearer and your recruitment strategy also.

1.3 Thank you for adding cluster characteristics to Table 1. Please can you add an indication of what is in parentheses in the table. I might assume SD, but it is helpful to know. It was interesting to see the differences that looked quite significant between clusters. OK

1.4. Re the flowchart, it would be helpful however, to have the number of eligible women in the I and C clusters to see how balanced they began. However, please include those missing for the primary outcome analysis – 311 to 275

1.5 I am now very confused about your power calculation. In the original paper you state you require 400 to achieve a moderate to small effect size. This reviewer thinks that for a paper to be useful, the power calculation should be verifiable and the description clear. So, you calculated 200 for each study condition (do you mean IPV and mental health or the two MH measures?). What was the effect size examined for each condition and is the MH effect clinically meaningful? Where did the relatively small ICCs come from - please provide a reference. What is impact on power of only having 43 groups rather than 63 clusters for your analysis? This response needs further revision, but I am glad you know cite your ICCs for future studies

1.6 Refs 25-27 are your AAS and mental health measures. Fortunately, in your text, you cite the correct numbers, but for references online, you should include the website and date accessed. Please make 29 and 30 more accessible please. These are international organisations with websites, and it is unclear where those you cite can be found. Needs correction

1.7 As the DHS measures are one of your primary outcomes, it would be useful to know which questions you used. The WHO MCS has many questions, as does the DHS. Which ones did you use, please add? If space is an issue, drop the AAS screening questions, as there are few of these - especially as IPV (DHS) were far more prevalent than your MH measures.

1.8 OK- most people use complete case analysis for ITT. OK

1.9 Good response. OK

1.10 OK thanks

1.11 OK

1.12 Definition helpful OK

1.13 IRB numbers added thx OK

1.14 It is important to see the n in your tables as it is smaller than the original sample. OK

1.15 Renumbered OK

1.16 OK

Discussion. Are the MH differences found clinically significant? Please comment.

Reviewer #3: The manuscript entitled ’Evaluation of an integrated intervention to reduce psychological distress and intimate partner violence in refugees: Results from the Nguvu cluster randomized feasibility trial’ with the aim to examine the relevance, acceptability, and feasibility of a multi-sectoral integrated violence and mental health-focused intervention (Nguvu).

This is an interesting study. The manuscript can be further improved based on the comments below.

Materials and Methods

Line 160, information on allocation of concealment, blinding, person who performed the randomization to be stated.

The language version used in the assessment/questionnaires to be stated.

Line 193-194, type of mental health outcomes, effect size figures, alpha to be stated and reference(s) to be cited.

Line 221-228, more information to be provided for the usual care group i.e what is happening between baseline to 9 weeks after receiving information at baseline.

For the low education level subjects, were there any difficulties in understanding the content of the study/questionnaires or questions asked?

Statistical analyses

Line 263, statistical tests/approach for each analysis to be clearly stated including statistical software which was used to derive the OR, 95%CI, effect size, correlation, ICC, regression analysis output.

Line 271, the sentence ‘To examine….’ to be placed in a new paragraph.

Line 274, the sentence requires revision.

Results

Table 1, the word 95% CI in the OR column to be removed.

Line 387-395, 401-403, 418-433 to be presented in table form.

Table 4, the word per protocol analysis to be omitted from model 4 and place on top of the first row along with intent to treat analysis. In the title, intent to treat and per protocol analysis to be stated. For the variable children, which variable children was used to be clearly stated/denoted.

For the convenient of reader, ensure the sequence of results presentation is aligned to the sequence stated in the statistical analysis section.

Figure 1, some figures not tally.

References to conform with the journal format.

Reviewer #4: This is a paper of a feasibility trial of a mental health and IPV intervention with refugees in Tanzania. The authors were responsive to comments of the previous reviewer. The authors did an excellent job describing the results of the trial. It would be helpful to add more information to the introduction to frame the significance and innovation of this intervention for this particular population and the theoretical framework that motivated the intervention. The paper would also benefit from adding some information to the methods section on the description of the intervention itself, so that the statements in the discussion are more easily interpretable. The discussion would benefit from describing what the researchers learned about conducting IPV/mental health interventions with refugees and what recommendations they have for future intervention research with refugees.

INTRODUCTION

Intro needs to discuss background on what are the specific mental health issues and IPV needs among refugees. How do refugees’ rates of mental health and IPV compare to the general population in Tanzania or DRC?

Have there been other interventions that have been tried with the same population? What did they find? How does this intervention compare with those?

p. 5, lines 82-85: This sentence could benefit from more clarity. Who is pathologizing women? Why do we need a critical feminist analysis? Any citations for this analysis and how it should be applied here?: “Studies that indicate poor mental health in the survivor may lead to increased risk for IPV require a critical feminist and human rights analysis to avoid pathologizing and blaming women for the crime of male-perpetrated violence in relationships.”

p. 5, line 88: women with mental disorders related to IPV – what are these specific disorders?

p. 6, lines 96-97: Need rationale for why this intervention is especially necessary in low-resource, refugee settings.

The third paragraph in the Intro (page 6) is about the formative research that lead to the Nguvu intervention. This feels like it should go in the Methods.

P. 6-7, lines 111-117: this is one very long sentence that is hard to follow. It should be broken up for readability.

p. 7: lines 126-130 This info is unclear and needs better explanation/elaboration for the reader to understand the importance of the study context, the multi-sectoral nature of the intervention… .

METHODS

Suggest new paragraph at line 169: “Eligible participants were adult (18+ years) female Congolese refugees living in…”

lines 187-189, this sentence is long and could use some editing to make it more succinct: “Women recruited from women’s groups randomized to Nguvu were enrolled in the intervention, whereas none of the women who were recruited from women’s groups randomized to the control received the intervention.

Was there a minimum threshold of number of members per group, in order to have the group included in the study? (for example, if only 2 members of a group participated in the study, did you retain the group in the study?)

Were all members of a group allocated to the same condition?

It would be useful to provide a short summary of the intervention in the Methods section. The authors provide citations for the intervention description, but given that this is a paper on the outcomes of the intervention, it would be useful to know some key details about the intervention here. For example, were the sessions delivered in person? Were the sessions designed to address some group-level variables, such as social cohesion? It is important to have these things in mind when evaluating the results.

Lines 225-226: What types of legal services were available to the women? What usual care services were available?

Lines 243-245: Why was IPV captured over the past two weeks and not the past month?

Lines 249-251: What is the intervention cohort study? What is the validation study? Are these the same thing?

Lines 255-257: How was acknowledging the sensitivity of the sexual violence questions supposed to improve the test-retest reliability? The connection between these two things is unclear.

Lines 269-270: Please explain what you mean by “We examined sensitivity to change in IPV… by calculating the change in primary and secondary outcomes”. What statistical analysis were you performing to examine “sensitivity”?

Lines 300-301: It sounds a bit odd to refer to documenting adverse events as an indicator of acceptability. Adverse events, on the face of it, don’t seem like the only variable for whether participants find the intervention acceptable. Perhaps framing adverse events as a safety indicator related to acceptability would help.

Lines 308-309: What types of deviations to the study protocol were recorded?

Please describe somewhere in the measures section: what is the significance of the participants’ resettlement process? What does this variable indicate?

RESULTS

Given that there were groups that were as small as 3 people, did that interfere with the way the intervention was intended? Did the analysis control for group size?

Lines 345-347: Why is the word “appeared” used in this sentence; was it higher or not? Were these means not compared statistically? If not, why?: “The proportion of women who were not married, but living with their partner, more highly educated, and literate appeared to be higher in women’s groups allocated to the Nguvu condition…”

Table 1:

- This table packs in a lot of information and it is a bit difficult to read. I suggest reducing the amount of text, if possible. For example, under the Resettlement category, I think it is not necessary to repeat “Respondent/household” for each value listed.

- Are the individual n (%) reported? There is text saying that they are listed for each category but I’m not sure which number they refer to. Can you just add a column? For example, “Education level, cluster mean %(SD), individual n(%)”

Table 2 is a nice summary of findings.

Table 3 – could you move the M(SD) and n(%) to column headings so that you don’t have to repeat them every row?

DISCUSSION

It would be helpful in the discussion to return to the very important issue of how this kind of an intervention should work in a refugee setting. Are there specific barriers or characteristics in the refugee setting that must be addressed in future interventions? What did you learn about this particular setting from this intervention?

You mention several key components of the intervention (task sharing, Cognitive Processing Therapy) that are mentioned for the first time in the discussion. It is hard to interpret these statements without reading some background on these components earlier in the paper.

7. PLOS authors have the option to publish the peer review history of their article (what does this mean?). If published, this will include your full peer review and any attached files.

Reviewer #1: No

Reviewer #3: No

Reviewer #4: **Yes: **Thespina J Yamanis

---

## [Author Response · Author response to Decision Letter 1]

30 Apr 2021

Reviewer #1

R1.1 Thank you for revising the flowchart table and recruitment description. Your description of the design of the study is much clearer and your recruitment strategy also.

Response: Thank you for your review of this revised draft of the manuscript. We believe that the paper is much stronger after considering your previous comments as well as the additional suggestions described below.

R1.2 Thank you for adding cluster characteristics to Table 1. Please can you add an indication of what is in parentheses in the table. I might assume SD, but it is helpful to know. It was interesting to see the differences that looked quite significant between clusters. 

Response: We have indicated whether we are reporting the mean and standard deviation as compared to the sample size and percentage in the first column next to each variable. We have also added a column header to clarify this point.

R1.3 Re the flowchart, it would be helpful however, to have the number of eligible women in the I and C clusters to see how balanced they began. However, please include those missing for the primary outcome analysis – 311 to 275

Response: Thank you for this suggestion. We added in the number of women in the women’s groups randomized to intervention and control. We added the full sample size and the number included in the follow-up to the figure title. This information is also included in the flow chart.

R1.4 I am now very confused about your power calculation. In the original paper you state you require 400 to achieve a moderate to small effect size. This reviewer thinks that for a paper to be useful, the power calculation should be verifiable and the description clear. So, you calculated 200 for each study condition (do you mean IPV and mental health or the two MH measures?). What was the effect size examined for each condition and is the MH effect clinically meaningful? Where did the relatively small ICCs come from - please provide a reference. What is impact on power of only having 43 groups rather than 63 clusters for your analysis? This response needs further revision, but I am glad you know cite your ICCs for future studies

Response: Thank you for noting the lack of clarity in our description of the power calculation. We have added more details to address the concerns noted by the reviewer.

“We planned to enroll 400 participants in this study, which would allow us to identify a small to moderate effect size for depression/anxiety measured using the average item score on the Hopkins Symptom Checklist (mean difference=1.6) and post-traumatic stress symptoms measured using the average item score on the Harvard Trauma Questionnaire (effect size=1.3). Enrolling 200 participants per study condition from the 63 original women’s groups with approximately equal allocation to each study condition was necessary to achieve at least 80% power to detect a small to moderate effect of the intervention on mental health outcomes accounting for up to 20% attrition and an intraclass correlation ranging from 0.1-0.5 within subjects and 0.1-0.3 within cluster[12]. Parameter estimates used to inform this power calculation were based on estimates from a previous cluster randomized controlled trial of Cognitive Processing Therapy conducted in the eastern Democratic Republic of the Congo that used the same outcome measures as were used in this feasibility trial[18].” - Pgs. 10-11 

R1.5 Refs 25-27 are your AAS and mental health measures. Fortunately, in your text, you cite the correct numbers, but for references online, you should include the website and date accessed. Please make 29 and 30 more accessible please. These are international organisations with websites, and it is unclear where those you cite can be found. Needs correction

Response: Thank you for this suggestion. We have added URLs and dates accessed to these references.

R1.6 As the DHS measures are one of your primary outcomes, it would be useful to know which questions you used. The WHO MCS has many questions, as does the DHS. Which ones did you use, please add? If space is an issue, drop the AAS screening questions, as there are few of these - especially as IPV (DHS) were far more prevalent than your MH measures.

Response: We have added more information about the DHS items that were used to assess IPV to the description of the measures.

“Psychological, physical, and sexual IPV were measured in the baseline and follow-up assessment using an adapted version of the Conflict Tactics Scales that was developed for the World Health Organization multi-country study on women’s health and domestic violence against women and the Demographic and Health Surveys [28-30]. The 11 Demographic and Health Survey items that were used include questions assessing lifetime and past two-week psychological violence (2 items: humiliate, threaten), physical violence (7 items: push/shake/throw, slap/twist arm/pull hair, punch, kick/drag, strangle/burn, threaten with weapon, attack with weapon), and sexual violence (2 items: forced sex, forced other sexual acts). IPV frequency was calculated as the mean reported frequency of each type of IPV (physical, psychological, sexual) over the past two-weeks. “ Pgs. 14-15

R1.7 Discussion. Are the MH differences found clinically significant? Please comment.

Response: The between-group differences at endline revealed small effect sizes that may not be clinically relevant. We appreciate this suggestion to describe this in our interpretation of the results and have added information on this point to the discussion. 

“It appeared that mental health was more sensitive to change, but between-group differences revealed small effect sizes that may not be clinically significant.” – Pg. 32

Reviewer #2:

No comments

Reviewer #3: 

R3.1 The manuscript entitled ’Evaluation of an integrated intervention to reduce psychological distress and intimate partner violence in refugees: Results from the Nguvu cluster randomized feasibility trial’ with the aim to examine the relevance, acceptability, and feasibility of a multi-sectoral integrated violence and mental health-focused intervention (Nguvu). This is an interesting study. The manuscript can be further improved based on the comments below.

Response: Thank you for your thoughtful comments on the manuscript. Please see below for a summary of the revisions we have made in response to these suggestions.

Materials and Methods

R3.2 Line 160, information on allocation of concealment, blinding, person who performed the randomization to be stated.

Response: Thank you for pointing out that this information is missing from the paper. We have added this information to the methods and also discuss some of the limitations related specifically to blinding/masking in the discussion.

“Women’s groups were allocated to the Nguvu intervention versus the usual care condition using a simple, unrestricted randomization approach with approximately equal allocation conducted by an investigator not affiliated with the current study[22]. The random number sequence was generated in Stata, Version 14[23]. All clusters were randomized at the same time, thus reducing concerns about allocation concealment[24].” – Pg. 9

“Participants and intervention facilitators were not masked to study allocation. While research assistants were not informed of the participants’ allocation, it may have become apparent through information shared during the endline assessment.” – Pg. 11

“It is also possible that the lack of masking of study participants may have influenced reporting. Furthermore, while the research assistants were not informed of the participants’ allocation, it is plausible that information divulged during the assessments may indicate whether they were participating in the Nguvu intervention, of which the research assistants were familiar and may have introduced biases in outcome assessment.” – Pg. 35 

R3.3 The language version used in the assessment/questionnaires to be stated.

Response: Thank you for noting this omission. We have added that “All measures were translated into Kiswahili…” to the description of the measures on Pg. 15

R3.4 Line 193-194, type of mental health outcomes, effect size figures, alpha to be stated and reference(s) to be cited.

Response: We have added more specific details about the mental health outcomes, effect sizes, and internal consistency to the methods section. We also added the internal consistency of each of the mental health outcomes and functional impairment to the results section.

“We planned to enroll 400 participants in this study, which would allow us to identify a small to moderate effect size for depression/anxiety measured using the average item score on the Hopkins Symptom Checklist (mean difference=1.6) and post-traumatic stress symptoms measured using the average item score on the Harvard Trauma Questionnaire (effect size=1.3). Enrolling 200 participants per study condition from the 63 original women’s groups with approximately equal allocation to each study condition was necessary to achieve at least 80% power to detect a small to moderate effect of the intervention on mental health outcomes accounting for up to 20% attrition and an intraclass correlation ranging from 0.1-0.5 within subjects and 0.1-0.3 within cluster[12]. Parameter estimates used to inform this power calculation were based on estimates from a previous cluster randomized controlled trial of Cognitive Processing Therapy conducted in the eastern Democratic Republic of the Congo that used the same outcome measures as were used in this feasibility trial[18].” - Pgs. 10-11 

“All primary outcome measures displayed good internal consistency at baseline (Anxiety: α=0.761, Depression: α=0.741, PTSD: α=0.722). These symptoms were higher among the usual care participants relative to participants in the Nguvu study condition. Functioning was similar between groups and reflected a little to moderate amount of difficulty completing common life tasks (α=0.841).” – Pg. 22

R3.5 Line 221-228, more information to be provided for the usual care group i.e what is happening between baseline to 9 weeks after receiving information at baseline.

Response: We have added more information about the services that were available in Nyarugusu to the description of the usual care condition.

“Women recruited from women’s groups randomized to the usual care condition received information about existing services for mental health and protection from the research assistant at the end of the baseline assessment. The gender-based violence response program that existed at the time of the study consisted of case management (including basic counseling) and referrals to protection, medical, or legal services. These services included legal consultation and aid services, education about women’s rights, and arranging safe shelter and accommodations[38].” - Pgs. 13-14

R3.6 For the low education level subjects, were there any difficulties in understanding the content of the study/questionnaires or questions asked?

Response: All measures underwent extensive pilot testing in the same population and setting. The measures were adapted so they were comprehensible as well as reliable and valid in the study population. We added a statement describing this to the description of the measures (Pg. 14) and included reference to the publication describing these measurement findings.

Statistical analyses

R3.7 Line 263, statistical tests/approach for each analysis to be clearly stated including statistical software which was used to derive the OR, 95%CI, effect size, correlation, ICC, regression analysis output.

Response: Thank you for this suggestion. We have added more detailed information about each of the statistical tests used to the statistical analysis section as well as the software used (Pgs. 16-18)

R3.8 Line 271, the sentence ‘To examine….’ to be placed in a new paragraph.

Response: We have reformatted this sentence so it now introduces a new paragraph (Pg. 16)

R3.9 Line 274, the sentence requires revision.

Response: Thanks for noting this grammatical error. We have revised this sentence as follows:

“Analyses compared outcomes between study conditions as they were assigned at randomization.” Pg. 16

Results

R3.10 Table 1, the word 95% CI in the OR column to be removed.

Response: We removed ‘95% CI’ from the OR column. Thank you.

R3.11 Line 387-395, 401-403, 418-433 to be presented in table form.

Response: Thank you for making this suggestion. We have added a supplemental file that includes the requested information in tabular format. 

R3.12 Table 4, the word per protocol analysis to be omitted from model 4 and place on top of the first row along with intent to treat analysis. In the title, intent to treat and per protocol analysis to be stated. For the variable children, which variable children was used to be clearly stated/denoted.

Response: We have revised table 4 according to the reviewer’s recommendations. These changes include reformatting the table make the intent-to-treat and per-protocol analysis columns consistent, adding ‘intent-to-treat’ and ‘per-protocol’ analysis to the title of the table, and clarifying that ‘number of children’ was the variable included in the adjusted models.

R3.13 For the convenient of reader, ensure the sequence of results presentation is aligned to the sequence stated in the statistical analysis section.

Response: We have moved the results related to acceptability and feasibility to follow all results related to relevance so that the results better align with the flow presented in the analysis section. Thank you for this suggestion.

R3.14 Figure 1, some figures not tally.

Response: We have double checked the numbers and believe the reviewer may be referencing the fact that the reasons for exclusion at the screening stage sum to a number greater than the number of people excluded. This is due to some participants meeting more than one criterion for exclusion. We have clarified this in the caption for the figure.

R3.15 References to conform with the journal format.

Response: Thank you. We have reviewed the references and have updated them according to the author guidelines for PLOS ONE.

Reviewer #4: 

R4.1 This is a paper of a feasibility trial of a mental health and IPV intervention with refugees in Tanzania. The authors were responsive to comments of the previous reviewer. The authors did an excellent job describing the results of the trial. It would be helpful to add more information to the introduction to frame the significance and innovation of this intervention for this particular population and the theoretical framework that motivated the intervention. The paper would also benefit from adding some information to the methods section on the description of the intervention itself, so that the statements in the discussion are more easily interpretable. The discussion would benefit from describing what the researchers learned about conducting IPV/mental health interventions with refugees and what recommendations they have for future intervention research with refugees.

Response: Thank you for your thorough review of the manuscript. We appreciate these insightful comments and have revised the paper to reflect your suggestions. Please see our specific revisions detailed in your points below. Overall, we added more information to the introduction to improve the framing of the paper as well as details of the intervention itself. We have added to the discussion as well, but as noted below, our findings regarding implementation are forthcoming in a paper detailing the results of the process evaluation and, unfortunately, out of scope for the current paper that focuses on the quantitative feasibility trial results.

INTRODUCTION

R4.2 Intro needs to discuss background on what are the specific mental health issues and IPV needs among refugees. How do refugees’ rates of mental health and IPV compare to the general population in Tanzania or DRC?

Response: We have added more specific statements about the prevalence of mental disorder among refugees to the introduction, while acknowledging the limited reliability of prevalence estimates for refugees globally. We have also added to the introduction reference to studies estimating the prevalence of common mental disorder in the eastern DRC and the region more broadly. Research indicates that there is an elevated burden of IPV and mental health problems among Congolese refugees in Tanzania and the region, but reliable epidemiological estimates are not available so we are unable to directly compare these estimate to those of the host population in Tanzania or in eastern DRC. We now briefly mention these limitations of existing data and have added references to available studies in the introduction.

“Recent estimates from the World Health Organization suggest that 22.1% of individuals in conflict-affected populations has a mental disorder at any point in time, which is approximately three times higher than non-conflict affected populations [2]. The burden of common mental disorder is estimated to be even greater among refugees and populations displaced by emergencies, but reliable epidemiological estimates are limited[3].” – Pg. 5

“Regionally, studies estimate that 39-44% of women in sub-Saharan Africa have experienced IPV in their lifetime[14, 15].” – Pg. 5 

“Nguvu, Kiswahili for strength and power, was developed among female refugees from the eastern Democratic Republic of the Congo (DRC). Research among women in the eastern DRC and refugees from this region consistently reports high levels of gender-based violence, particularly IPV (e.g., 31% of adult women in the eastern DRC), as well as co-occurring mental health problems in survivors of IPV [16]. However, the reliability and availability of prevalence estimates among refugee populations specifically are limited. One population-based survey conducted in the eastern DRC found that experiencing IPV is associated with a higher probability of common mental health problems including depression (64.9% vs. 31.0%), post-traumatic stress disorder (77.2% vs. 43.9%), and suicidal ideation (42.4% vs. 20.5%) or attempts (33.1% vs. 9.7%)[17].” – Pgs. 6-7

R4.3 Have there been other interventions that have been tried with the same population? What did they find? How does this intervention compare with those?

Response: We are not aware of other comparable interventions that have been tested in the same population. This intervention draws from evidence produced by a study conducted in the eastern DRC focused on Cognitive Processing Therapy that found a longer version of the intervention to be associated with reductions in mental health problems among survivors of gender-based violence (Pg. 7). We have added a statement about this gap to the introduction.

“An integrated IPV and mental health intervention fills critical gaps in research and programming in refugee settings[20]. Very little is known about the strategies for reducing IPV among women experiencing ongoing violence in refugee settings and humanitarian emergencies[18].” – Pg. 7

R4.4 p. 5, lines 82-85: This sentence could benefit from more clarity. Who is pathologizing women? Why do we need a critical feminist analysis? Any citations for this analysis and how it should be applied here?: “Studies that indicate poor mental health in the survivor may lead to increased risk for IPV require a critical feminist and human rights analysis to avoid pathologizing and blaming women for the crime of male-perpetrated violence in relationships.”

Response: We have revised this sentence to improve clarity and added a recently published article that provides much more detail about why critical feminist theory and human rights are essential perspectives when interpreting the relationship between mental health and IPV. Thank you for this suggestion.

“These findings should be interpreted using a critical feminist and human rights analysis to avoid pathologizing and blaming women for the crime of male-perpetrated violence in relationships[18, 19].” – Pg. 5

R4.5 p. 5, line 88: women with mental disorders related to IPV – what are these specific disorders?

Response: We have added the specific mental disorders we were references to this sentence. 

“Further, women with common mental health conditions related to IPV (e.g, depression, anxiety, post-traumatic stress disorder) may also suffer from low self-esteem, self-blame, poverty and alcohol or drug use, factors that may reduce their capacity to escape from future victimization.” – Pg. 6

R4.6 p. 6, lines 96-97: Need rationale for why this intervention is especially necessary in low-resource, refugee settings.

Response: Thank you for this comment. To reinforce this point, we have added more detail about the burden of these co-occurring problems of IPV and psychological distress (see response to R4.2), a statement about the need to address these related public health priorities (see below), and mentioned that this fills an existing gap in programming and evidence. 

“The complex relationship between IPV and mental health therefore indicates the need for an integrated response to effectively address the dual priorities for improved mental health and reduction in IPV in global public health and human rights.” – Pg. 6

“It is important to initiate operational research to bridge the gap that often exists in humanitarian settings between mental health interventions that are usually not tailored to the needs of IPV survivors and routine IPV interventions that often lack the clinical competency to effectively address co-occurring mental health issues[22]. Rigorous research on strategies for reducing IPV among women experiencing ongoing violence in refugee settings and humanitarian emergencies is limited[20].” – Pg. 7

R4.7 The third paragraph in the Intro (page 6) is about the formative research that lead to the Nguvu intervention. This feels like it should go in the Methods.

Response: We have moved this paragraph to the measures under the ‘Intervention and Usual Care Condition’ section.

R4.8 P. 6-7, lines 111-117: this is one very long sentence that is hard to follow. It should be broken up for readability.

Response: Thank you for this recommendation. We have modified the original sentence to improve readability.

“Using information from these qualitative interviews along with expert consultation and a desk review [16], we utilized a modified six-session version of Cognitive Processing Therapy (CPT), an evidence-based intervention developed for survivors of assault [22, 23]. The twelve-session version of CPT has been shown to reduce mental health problems among Congolese survivors of gender-based violence in the DRC [24]. In the Nguvu intervention, CPT was combined with two sessions of advocacy counseling, which has demonstrated some reductions in IPV victimization among women [25-27].” – Pg. 13

R4.9 p. 7: lines 126-130 This info is unclear and needs better explanation/elaboration for the reader to understand the importance of the study context, the multi-sectoral nature of the intervention… .

Response: We have added more explanation about the why these contextual factors require different approaches to implementation

“These considerations include: 1) differences in the study context (e.g., instability and high rates of in- and out-migration resulting in challenges retaining participants in long interventions), 2) health providers (e.g., task-shifting with refugee incentive workers to leverage knowledge and trust, while overcoming the limited human resource capacity to provide mental health and IPV response services), 3) the multi-sectoral nature of the intervention positioned within humanitarian health and protection systems that have different mandates and are often implemented by different agencies, 4) challenges with communication and coordination across sectors, and 5) other challenges with the broader service delivery system (e.g., high rates of staff turnover) [20].” – Pg. 7

METHODS

R4.10 Suggest new paragraph at line 169: “Eligible participants were adult (18+ years) female Congolese refugees living in…”

Response: We have added a new paragraph starting with the sentence mentioned above.

R4.11 lines 187-189, this sentence is long and could use some editing to make it more succinct: “Women recruited from women’s groups randomized to Nguvu were enrolled in the intervention, whereas none of the women who were recruited from women’s groups randomized to the control received the intervention.

Response: We have revised this sentence as follows:

“Women recruited from women’s groups randomized to Nguvu were enrolled in the intervention. Women who were recruited from women’s groups randomized to the control condition received information about available protection and mental health services in Nyarugusu.” – Pg. 10

R4.12 Was there a minimum threshold of number of members per group, in order to have the group included in the study? (for example, if only 2 members of a group participated in the study, did you retain the group in the study?)

Response: Group sessions began when at least 6 women were enrolled and assigned to a given Nguvu group/facilitator pair. We did not exclude any groups from the analysis. We have added this information to the description of the intervention.

“Nguvu group sessions began when at least 6 women were enrolled and allocated to a given group.” – Pgs. 11-12

R4.13 Were all members of a group allocated to the same condition?

Response: Yes, all women from the same women’s group were allocated to the same condition given that randomization occurred at the cluster (i.e., women’s group) level.

“Women recruited from women’s groups randomized to Nguvu were enrolled in the intervention. Women who were recruited from women’s groups randomized to the control condition received information about available protection and mental health services in Nyarugusu.” – Pg. 12

R4.14 It would be useful to provide a short summary of the intervention in the Methods section. The authors provide citations for the intervention description, but given that this is a paper on the outcomes of the intervention, it would be useful to know some key details about the intervention here. For example, were the sessions delivered in person? Were the sessions designed to address some group-level variables, such as social cohesion? It is important to have these things in mind when evaluating the results.

Response: Thanks for this suggestion. We agree and have moved and added details of the intervention to the methods under the ‘Intervention and Usual Care Conditions’ section. We have added details about implementation that were missing in the original version (e.g., all sessions were delivered in person). We have also added details about the hypothesized mediation pathways (e.g., improved social support, coping). See Pgs. 11-13. 

R4.15 Lines 225-226: What types of legal services were available to the women? What usual care services were available?

Response: We have added a sentence about what types of services were available for women in Nyarugusu.

“The gender-based violence response program that existed at the time of the study consisted of case management (including basic counseling) and referrals to protection, medical, or legal services. These services included legal consultation and aid services, education about women’s rights, and arranging safe shelter and accommodations.” – Pgs. 13-14

R4.16 Lines 243-245: Why was IPV captured over the past two weeks and not the past month?

Response: We selected a shorter recall period to allow the measures to be more sensitive to change. Having a longer recall period may prevent us from detecting meaningful changes in study outcomes that would accrue over several intervention sessions. Furthermore, this recall period is consistent with the mental health outcome measures we used in this study.

R4.17 Lines 249-251: What is the intervention cohort study? What is the validation study? Are these the same thing?

Response: Yes, thank you for noting this point of confusion. We have replaced reference to the ‘validation study’ with ‘intervention cohort study’ to be consistent. We have provided a more detailed explanation of the purpose of the intervention cohort study in the methods.

“Details of the development and preliminary testing of the intervention through a non-controlled intervention cohort study that was previously conducted to pilot test the intervention and assess the psychometric properties of the study outcome measures are reported elsewhere [20].” – Pg. 13

R4.18 Lines 255-257: How was acknowledging the sensitivity of the sexual violence questions supposed to improve the test-retest reliability? The connection between these two things is unclear.

Response: In the intervention cohort study we identified poor test-retest reliability and, through qualitative interviews and discussion with our partners in Nyarugusu, we hypothesized that this was partially due to lack of trust and concerns about the consequences of reporting sexual violence. For test-retest reliability, this may have resulted in withholding this information during the initial assessment and, after realizing that the information shared during these interviews remained confidential and building rapport/trust with the research team, feeling more comfortable reporting sensitive information during the second interview. We have provided some additional information in the methods on this as well as reference to the intervention cohort study paper.

“We therefore made adaptations for the current study by adding a script that was read by the interviewer prior to administering the sexual violence items to acknowledge the sensitivity of these questions, while reassuring the participant that her answers would be kept confidential and there would not be any consequences of her reporting these experiences.” – Pg. 15

R4.19 Lines 269-270: Please explain what you mean by “We examined sensitivity to change in IPV… by calculating the change in primary and secondary outcomes”. What statistical analysis were you performing to examine “sensitivity”?

Response: Sensitivity to change analyses are largely descriptive. We calculated the mean change from baseline to follow-up and translated these into standardized effect sizes. We also examined whether these changes were correlated in expected directions as indications of the measures being suitable for primary outcome measures in a definitive RCT. We have added information to the text to further explain these analyses.

“We examined sensitivity to change in IPV, psychological distress, and functional impairment measures by calculating the mean change in primary and secondary outcomes from baseline to follow-up, the effect size, as well as the correlation between these change scores. This analysis was conducted to determine whether these assessment tools would be suitable as primary outcome measures in a fully powered, definitive randomized controlled trial.” – Pg. 16

R4.20 Lines 300-301: It sounds a bit odd to refer to documenting adverse events as an indicator of acceptability. Adverse events, on the face of it, don’t seem like the only variable for whether participants find the intervention acceptable. Perhaps framing adverse events as a safety indicator related to acceptability would help.

Response: Thanks, we agree and have revised the sentence to frame it as recommended by the reviewer.

“To examine safety of the intervention, an indicator of acceptability, we documented adverse events.” – Pg. 18

R4.21 Lines 308-309: What types of deviations to the study protocol were recorded?

Response: We have added a summary of the type of deviations that were recorded.

“To evaluate feasibility of conducting a definitive randomized trial of Nguvu we monitored and described any deviations to the study protocol, including any changes to implementation of the recruitment, screening, assessment, or intervention procedures.” – Pg. 18

R4.22 Please describe somewhere in the measures section: what is the significance of the participants’ resettlement process? What does this variable indicate?

Response: At the time of the study there were ongoing resettlement activities in Nyarugusu. Given the salience of the resettlement process to the refugee experience and the implications of resettlement on retention, we included this question in assessments. We’ve added a brief description of this in our methods section.

“In addition to these outcome measures, we assessed demographic characteristics of the sample. Given ongoing resettlement efforts at the time of the study we assessed participant preferences toward resettlement and whether they had begun the process.” – Pg. 14 

RESULTS

R4.23 Given that there were groups that were as small as 3 people, did that interfere with the way the intervention was intended? Did the analysis control for group size?

Response: Thank you for raising this question. Women recruited from multiple women’s groups could be included in the same Nguvu group. Thus, as stated in our response to R4.12, Nguvu intervention groups had a minimum of 6 participants, which may include women recruited from one or more women’s groups. We have a sentence in the results describing the Nguvu group size and number of groups implemented as part of the study. We have reviewed the manuscript and tried to be explicit and clear when referring to the ‘women’s groups’ from which we recruited and the ‘Nguvu groups’ that women were assigned to for the intervention.

“Fifteen Nguvu groups that included 6-13 participants each were assembled and implemented throughout the course of the study period.” – Pg. 30

R4.24 Lines 345-347: Why is the word “appeared” used in this sentence; was it higher or not? Were these means not compared statistically? If not, why?: “The proportion of women who were not married, but living with their partner, more highly educated, and literate appeared to be higher in women’s groups allocated to the Nguvu condition…”

Response: Yes, you are correct. We have restated the sentence to more accurately describe the differences in baseline characteristics between study conditions.

“The proportion of women who were not married, but living with their partner, more highly educated, and literate was higher in women’s groups allocated to the Nguvu relative to usual care condition.” – Pg. 20

Table 1:

R4.25 This table packs in a lot of information and it is a bit difficult to read. I suggest reducing the amount of text, if possible. For example, under the Resettlement category, I think it is not necessary to repeat “Respondent/household” for each value listed.

Response: Great suggestion, thank you. We agree that Table 1 is quite dense and have followed your suggestions to reduce some of the words, especially under the ‘resettlement’ heading. We added information to characterize the clusters as recommended by a previous reviewer so would prefer not to eliminate any of the data in the table, but think your suggestion has improved the readability of the able. Thanks. 

R4.26 Are the individual n (%) reported? There is text saying that they are listed for each category but I’m not sure which number they refer to. Can you just add a column? For example, “Education level, cluster mean %(SD), individual n(%)”

Response: We have tried to clarify what is being reported in table 1 in a few ways. This is a bit complicated because the values characterizing the full sample, the clusters, and the individual participants. I have added a footnote to the table clarifying exactly what is reported in each column by variable type as well as a header in column 1. Hopefully this clarifies what information is being reported without adding too much text to the table

R4.27 Table 2 is a nice summary of findings.

Response: Thank you!

R4.28 Table 3 – could you move the M(SD) and n(%) to column headings so that you don’t have to repeat them every row?

Response: This is a very helpful suggestion. We have reformatted the table and reported the M(SD) and n(%) in the column headings as appropriate for each section of the table.

DISCUSSION

R4.29 It would be helpful in the discussion to return to the very important issue of how this kind of an intervention should work in a refugee setting. Are there specific barriers or characteristics in the refugee setting that must be addressed in future interventions? What did you learn about this particular setting from this intervention?

Response: This is a very important point. We appreciate the reviewer for raising it. The scope of this specific paper focuses on the quantitative results from the feasibility trial. In the discussion we reference a forthcoming publication that examines these implementation challenges and considerations in detail based on the results of a process evaluation we conducted after completion of this feasibility trial. While we agree that these are very important issues to discuss, they are outside the scope of this paper. We hope these results will be published and available soon.

R4.30 You mention several key components of the intervention (task sharing, Cognitive Processing Therapy) that are mentioned for the first time in the discussion. It is hard to interpret these statements without reading some background on these components earlier in the paper.

Response: Thank you for mentioning this. We have added a more detailed description of Cognitive Processing Therapy to the methods (under ‘Intervention and Usual Care Conditions’; Pgs. 11-13). We have also added in details explaining what we mean by task shifting as well as some references to support these statements (see below).

“Intervention facilitators were lay refugee incentive workers in Nyarugusu already working with the humanitarian partner (International Rescue Committee) who had some experience working with protection and psychosocial support programs in the camp and had received training from experts in trauma-informed psychological and gender-based violence interventions. Incentive workers are refugees who undertake work related to the provision of humanitarian assistance and receive fixed compensation referred to as an ‘incentive’ [28]. Consistent with task-shifting approaches[29], these incentive workers were non-specialists and had no prior experience in implementing psychological interventions beyond the basic psychosocial support programs offered by the humanitarian partner” – Pg. 12

---

## [Editor Report · Decision Letter 2]

27 May 2021

Evaluation of an integrated intervention to reduce psychological distress and intimate partner violence in refugees: Results from the Nguvu cluster randomized feasibility trial

PONE-D-20-13577R2

Dear Dr. Greene,

We’re pleased to inform you that your manuscript has been judged scientifically suitable for publication and will be formally accepted for publication once it meets all outstanding technical requirements.

Kind regards,

Michelle L. Munro-Kramer, PhD, CNM, FNP-BC

Academic Editor

PLOS ONE

Additional Editor Comments (optional):

Thank you for the thoughtful edits in response to the reviewer comments. We are happy to accept this manuscript for publication.
---

## [Editor Report · Acceptance letter]

11 Jun 2021

PONE-D-20-13577R2 

Evaluation of an integrated intervention to reduce psychological distress and intimate partner violence in refugees: Results from the Nguvu cluster randomized feasibility trial 

Dear Dr. Greene:

I'm pleased to inform you that your manuscript has been deemed suitable for publication in PLOS ONE. Congratulations! Your manuscript is now with our production department. 

Kind regards, 

on behalf of

Dr. Michelle L. Munro-Kramer 

Academic Editor

PLOS ONE